# Dynamic inter-domain transformations mediate the allosteric regulation of human 5, 10-methylenetetrahydrofolate reductase

Linnea K. M. Blomgren [1], Melanie Huber[1], Sabrina R. Mackinnon [2], Céline Bürer[1], Arnaud Baslé [2], Wyatt W. Yue [2,3,4] ✉, D. Sean Froese [1,4] ✉ & Thomas J. McCorvie [2,4] ✉

5,10-methylenetetrahydrofolate reductase (MTHFR) commits folate-derived one-carbon units to generate the methyl-donor s-adenosyl-L-methionine (SAM). Eukaryotic MTHFR appends to the well-conserved catalytic domain (CD) a unique regulatory domain (RD) that confers feedback inhibition by SAM. Here we determine the cryo-electron microscopy structures of human MTHFR bound to SAM and its demethylated product s-adenosyl-L-homo-cysteine (SAH). In the active state, with the RD bound to a single SAH, the CD is flexible and exposes its active site for catalysis. However, in the inhibited state the RD pocket is remodelled, exposing a second SAM-binding site that was previously occluded. Dual-SAM bound MTHFR demonstrates a substantially rearranged inter-domain linker that reorients the CD, inserts a loop into the active site, positions Tyr404 to bind the cofactor FAD, and blocks substrate access. Our data therefore explain the long-distance regulatory mechanism of MTHFR inhibition, underpinned by the transition between dual-SAM and single-SAH binding in response to cellular methylation status.

S-adenosyl-L-methionine (SAM) is the most important methyl group donor of the cell. It facilitates over 200 transmethylation reactions that modify DNA, RNA, proteins, and metabolites[1]. Indeed, SAM is the most widely used cellular co-substrate after ATP, and the ratio of SAM to its demethylated product s-adenosyl-l-homocysteine (SAH) serves as an index of the cell's methylation potential[2], driving the myriad of trans-methylation reactions in cells and tissues. In humans, SAM synthesis proceeds through adenosylation of methionine, whose precursor is homocysteine. This process requires the folate derivative 5-methyltetrahydrofolate (CH$_3$-THF), generated solely by the enzyme 5,10-methylenetetrahydrofolate reductase (MTHFR) from 5,10-methy-lenetetrahydrofolate (CH$_2$-THF)[3]. In so doing, MTHFR represents the first committed step to SAM biosynthesis, shuttling one-carbon units in the form of CH$_2$-THF away from nucleotide synthesis. MTHFR

dysfunction therefore has an enormous impact in human health, and has been associated with cancer[4,5], cardiovascular disease[6,7], and neural tube defects[8,9]. Notably, severe MTHFR deficiency, due to inherited bi-allelic disease-causing variants in the *MTHFR* gene, is a rare autosomal recessive disorder resulting from a loss of enzymatic function, which leads to homocystinuria and can be fatal if left untreated[10].

MTHFR can be found in all domains of life. In contrast to pro-karyotes, eukaryotes have evolved an elegant feedback regulation mechanism to modulate MTHFR activity. Human MTHFR is exquisitely sensitive to the SAM:SAH ratio, being allosterically inhibited by SAM[11,12] and re-activated by SAH[13,14]. The sensitivity of MTHFR towards SAM inhibition is also modulated by multiple Ser/Thr phosphorylations[15], whereby phosphorylated MTHFR has a reduced inhibition constant ($K_i$) for SAM and co-purifies binding a mixture of 0−2 SAM molecules

[1]Division of Metabolism and Children's Research Center, University Children's Hospital Zürich, University of Zürich, Zürich CH-8032, Switzerland. [2]Biosciences Institute, The Medical School, Newcastle University, Newcastle upon Tyne NE2 4HH, UK. [3]Centre for Medicines Discovery, Nuffield Department of Clinical Medicine, University of Oxford, Oxford OX3 7DQ, UK. [4]These authors contributed equally: Wyatt W. Yue, D. Sean Froese, Thomas J. McCorvie. ✉e-mail: Wyatt.Yue@newcastle.ac.uk; Sean.Froese@kispi.uzh.ch; Thomas.McCorvie@newcastle.ac.uk

per protein unit[16]. Conversely, when MTHFR is de-phosphorylated, it co-purifies while binding 0–2 SAH molecules[16].

Our recently determined crystal structure of as purified human MTHFR, in complex with SAH, revealed a two-domain enzyme architecture[16]. The catalytic domain (CD), composed of a TIM-barrel fold conserved across all species[16], binds the cofactor flavin adenine dinucleotide (FAD), the electron donor nicotinamide adenine dinucleotide phosphate (NADPH) and the substrate $CH_2$-THF, such that physiologically the forward activity leading to $CH_3$-THF formation takes place[17,18]. This CD is appended to a structurally unique regulatory domain (RD) via an extended inter-domain linker. The RD contains the binding pocket for SAH, and expectedly for SAM also. Combined crystallographic, small angle X-ray scattering (SAXS), and native mass spectrometry studies revealed significant intrinsic flexibility, particularly with respect to inter-domain orientations[16]. Such conformational flexibility of MTHFR is consistent with its complex regulatory mechanism, which remains to be characterised in molecular detail.

The catalytic mechanism of the MTHFR reaction towards forming $CH_3$-THF is well understood, through the plethora of structural and enzymatic studies from diverse orthologous proteins[3,19–22]; however, little is known about how this activity is allosterically regulated in eukaryotic enzymes. The binding of SAM and SAH to the RD have been suggested to induce different MTHFR conformers that correspond to the inhibited and dis-inhibited states, respectively[16]. Nevertheless, how this is related to the dimeric structure of MTHFR and the signal transduction within the monomers has not yet been characterised.

In this work, we have subjected full-length human MTHFR incubated with SAM or SAH to cryo-electron microscopy (cryo-EM). The multiple structures determined provide insights into the large conformational changes necessary for MTHFR allosteric regulation. In combination with functional assays, our structures present an unexpected discrepancy in the binding stoichiometry of SAM and SAH, and detail how SAM binding remodels the allosteric pocket and induces structural rearrangements of the inter-domain linker that are key to mediating the conformational states of MTHFR.

## Results

### Inter-domain rearrangement of MTHFR in response to SAM/SAH

The crystal structure of human MTHFR[16], derived from a non-phosphorylatable truncation construct (MTHFR$_{trunc}$ aa 38–644), may not reflect the full capability of the protein in its conformational dynamics during catalysis and regulation. Therefore, we pursued structural studies using the full-length protein (MTHFR$_{FL}$ aa 1–656) that harbours the N-terminal Ser/Thr-rich region and is multiply phosphorylated during its recombinant expression (Fig. 1a). Initial negative stain electron microscopy revealed 2D classes consisting of four or three lobes. Two distinct types of classes were present when the protein was incubated with either SAH or SAM (Supplementary Fig. 1a, b). One type of 2D classes showed an extended conformation of the MTHFR homodimer, possibly corresponding to the MTHFR$_{trunc}$ crystal structure complexed with SAH that triggered a dis-inhibited state of the enzyme (MTHFR$_{trunc}^{SAH}$, PDB: 6fcx)[16]. The other type of 2D classes exhibited a more compact conformation and appeared to be dimeric MTHFR with different domain orientations, possibly representing an inhibited state of MTHFR triggered by the presence of SAM.

Prompted by the possibility to investigate both SAM- and SAH-induced conformers from the full-length protein sample, we proceeded with cryo-EM data collections to gain higher resolution structures. MTHFR$_{FL}$ protein suffered from orientation bias on grids, which we partially overcame by the application of C2 symmetry in refinements (Supplementary Table 1 and Supplementary Figs. 2a–h, 3a–e, 4a–c and 5a–e). We also combined particles from different grids to diversify any captured views as different grids had different preferred orientations[23] (Supplementary Figs. 2a, b, 3a, b and 6a–f). A combination of global and masked local refinements with symmetry expansion

resulted in three maps ranging from 2.8 to 3.1 Å resolution that allowed us to confidently model both the SAH- and SAM-bound states of MTHFR.

In the presence of excess SAH (~100-fold relative to protein), a concentration at which MTHFR is fully active (Supplementary Fig. 7a), a masked local refinement at 2.8 Å resolution allowed a reconstruction (MTHFR$_{FL}^{SAH\,(symm)}$) of the MTHFR homodimeric interface involving the two RDs. This is highly analogous to the SAH-bound MTHFR$_{trunc}^{SAH}$ crystal structure ($C^\alpha$-RMSD 0.536 Å), with respect to the dimer packing arrangement, as well as the placement and orientation of one SAH molecule within the RD pocket (Supplementary Figs. 2a–e and 7b). Beyond the RD, however, there is high flexibility in the linker and CD regions in our locally refined map. This is strongly reflected in an asymmetric map at 3.1 Å resolution (MTHFR$_{FL}^{SAH\,(asymm)}$) obtained from symmetry expansion and 3D classification (Fig. 1b and Supplementary Fig. 2a, b, f–h). This asymmetry was also seen in the ab initio map during initial processing (Supplementary Fig. 2b). Here, the MTHFR homodimers appeared as asymmetric particles, where for one protomer both the linker and CD are highly disordered. In contrast, density was present for the linker and CD of the other protomer, though the density of both suggested flexibility (Fig. 1b, c and Supplementary Fig. 7c). It is important to note that the preferential orientation and possible denaturation at the air-water interface may explain the missing CD. Therefore, due to these issues, our modelling of this map is biased towards our previously determined crystal structure of MTHFR$_{trunc}^{SAH}$ (see Methods). This SAH-bound crystal structure shows asymmetry in terms of the CDs where one is partially disordered and more flexible than the other[16]. SAXS also suggested that SAH-bound MTHFR is more flexible in solution likely due to flexibility of the linkers and CDs[16]. This flexibility, the approximate CD-RD orientation, and conformational asymmetry seen in our model agrees with that of the SAH-bound MTHFR$_{trunc}^{SAH}$ crystal structure ($C^\alpha$-RMSD 0.826 Å) (Supplementary Fig. 7b). All considered together, our findings are consistent with the notion that SAH renders the MTHFR protein to be active, whereby the CDs are highly flexible and provide unhindered access of the substrate $CH_2$-THF and electron donor NADPH to the active site for catalysis.

Following incubation with 500-fold excess SAM, a ligand concentration where MTHFR is fully inhibited (Supplementary Fig. 7a), we determined one map at 2.9 Å resolution with C2 symmetry applied (Fig. 1b and Supplementary Fig. 3a–e). The high quality of the map allowed us to model the bulk of the CD (aa 44–160, 171–341), part of the linker region (380–412), and the near entirety of the RD (aa 413–644) (Fig. 1a, b). It is immediately apparent that this SAM-induced structure (MTHFR$_{FL}^{SAM}$) adopts drastically different inter-domain orientations compared to the SAH-bound cryo-EM (MTHFR$_{FL}^{SAH\,(asymm)}$, this study) and the crystal[16] structure (MTHFR$_{trunc}^{SAH}$)[16]. By overlaying the MTHFR$_{FL}^{SAM}$ (cryo-EM) and MTHFR$_{trunc}^{SAH}$ (crystal) structures using the homodimeric RD interface ($C^\alpha$-RMSD 0.691 Å) (Supplementary Fig. 7d), we observed a significant rigid body movement of each CD relative to its own RD (Fig. 1b). This inter-domain rearrangement manifests as a ~34° rotation coupled to ~14.8 Å translation, along an axis that intersects with the linker region connecting the CD and RD (Supplementary Fig. 7e). Such a conformational change is consistent with our negative stain 2D classes (Supplementary Fig. 1a, b) and is reflected by both FAD emission and differential scanning fluorimetry (DSF) experiments. Here 1 mM SAM resulted in a 3-4 °C increase in the melting temperature of as-purified MTHFR, whereas incubation with SAH resulted in no increase in stability (Supplementary Figs. 7f and 8a–d). This two-state transition, which essentially displaces the CD active site by nearly 30 Å (Fig. 1c), can be likened to the opening and closing of an oyster shell that controls the accessibility of its shell content (i.e. the active site) (Fig. 1d). As a result, the CD rotation narrows the cleft between the domains and traps the linker between them.

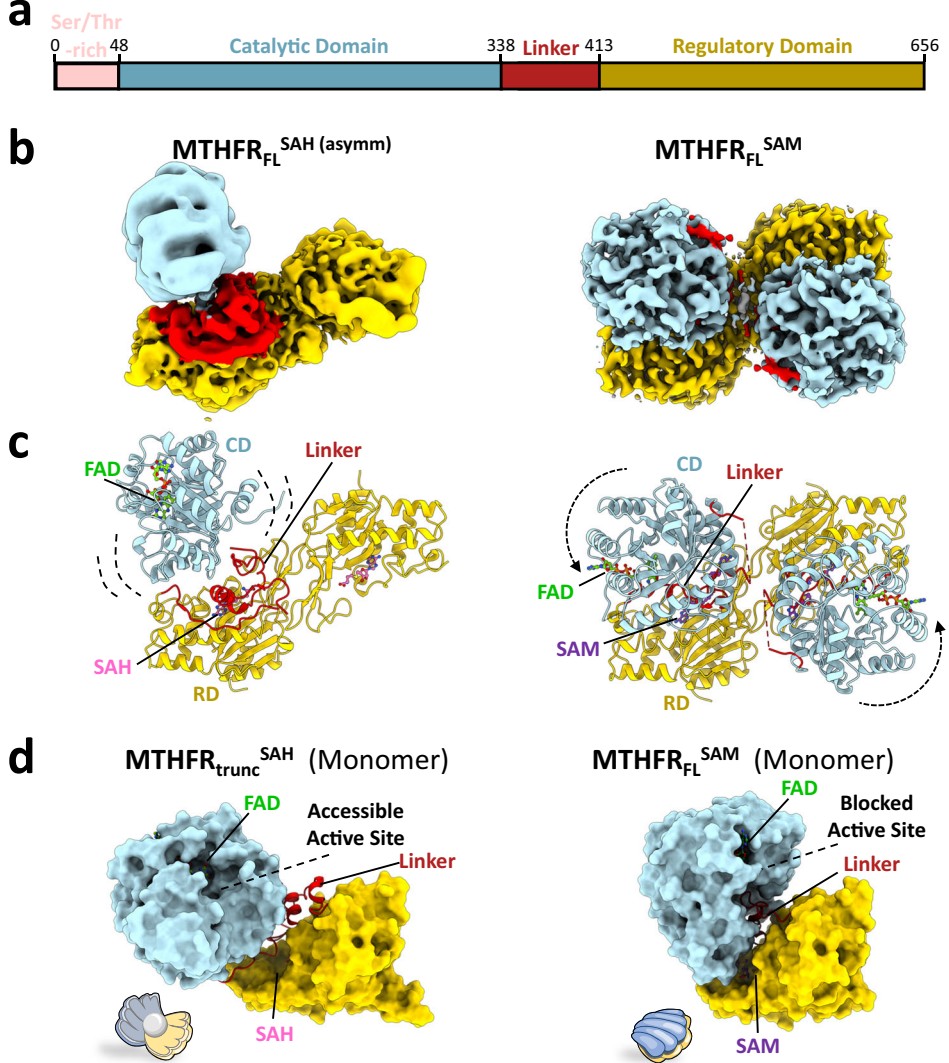

**Fig. 1 | Conformational change from open SAH-bound to closed SAM-bound MTHFR$_{FL}$. a** Schematic representation of full-length human MTHFR$_{FL}$, used for cryo-EM experiments and coloured according to protein domains. The colour coding is maintained in subsequent structural representations. Cryo-EM maps (**b**) and corresponding molecular models (**c**) of MTHFR$_{FL}^{SAH (asymm)}$ (left) and MTHFR$_{FL}^{SAM}$ (right). Note the flexibility of the CD in MTHFR$_{FL}^{SAH (asymm)}$ (represented by dashed brackets) and the rigid body displacement and rotation of the CD with respect to the rigid RD in MTHFR$_{FL}^{SAM}$ (represented by dashed arrows). Identified ligands are indicated in **c. d** Surface representation of monomeric MTHFR$_{trunc}^{SAH}$ (left), with accessible active site resembling an open oyster shell, and monomeric MTHFR$_{FL}^{SAM}$ (right) with a closed inter-domain arrangement and limited access to the active site resembling a closed oyster shell.

## Steric blockage of the active site from linker insertion causes allosteric inhibition

MTHFR across all domains of life retains a common binding site for the electron donor NAD(P)H and substrate CH$_2$-THF at the entrance of the CD active site[16,19,20,24]. As part of the bi-bi ping-pong catalytic mechanism[24,25], NAD(P)H (Fig. 2a) and CH$_2$-THF (Fig. 2b) stack against the flavin cofactor FAD in turn to carry out hydride transfer reactions. In our MTHFR$_{FL}^{SAM}$ structure, FAD was found in both CD active sites of the homodimer (Fig. 2c) with clear electron densities (Supplementary Fig. 9a), which superimpose well with those found in MTHFR$_{trunc}^{SAH}$ and orthologue structures (Supplementary Fig. 9b).

Notably, the direct consequence of the inter-domain rearrangement in the presence of SAM is that the CD active site is now oriented towards a different environment of the linker and RD regions (Fig. 2d). To our surprise, the entrance to the CD active site of MTHFR$_{FL}^{SAM}$ is blocked by a hydrophobic insertion of the linker region, bringing in three aromatic residues from a Y$_{403}$YLF$_{406}$ sequence (Tyr403, Tyr404, Phe406) to fill the space where NADPH/CH$_2$-THF is expected to bind (Fig. 2c). In addition to occluding the active site entrance, the linker

residue Tyr404 mimics NADPH and CH$_2$-THF through π-π stacking with the FAD isoalloxazine ring, in addition to interacting with Glu63, Thr94, and Tyr321 (Supplementary Fig. 9b) – strongly/strictly conserved residues across all kingdoms of life[16] – that are involved in CH$_2$-THF and NAD(P)H binding (Fig. 2a, b and Supplementary Fig. 9b). The hydrophobic insertion is further accommodated in the CD through a reorientation of helix α8, which flanks the NADPH/CH$_2$-THF binding site. Here, helix α8 is displaced away from its position in MTHFR$_{trunc}^{SAH}$ and re-oriented towards the exterior (Fig. 2c and Supplementary Fig. 9c). As a result, residues Lys270, Leu271, and Ser272 of helix α8, which are conserved from mammals to worms (Supplementary Fig. 10) and previously implicated in differentiating specificity between NADH (electron donor in bacterial MTHFRs)[24,26] and NADPH (electron donor in eukaryotic MTHFRs)[16], have been displaced by ~7 Å to make way for the hydrophobic insertion. The sidechain of Lys273, which would otherwise interact with CH$_2$-THF/NADPH, is also directed away from the binding site (Supplementary Fig. 9c).

The hydrophobic Y$_{403}$YLF$_{406}$ insertion from the linker therefore represents an auto-inhibitory element sterically blocking the NADPH/

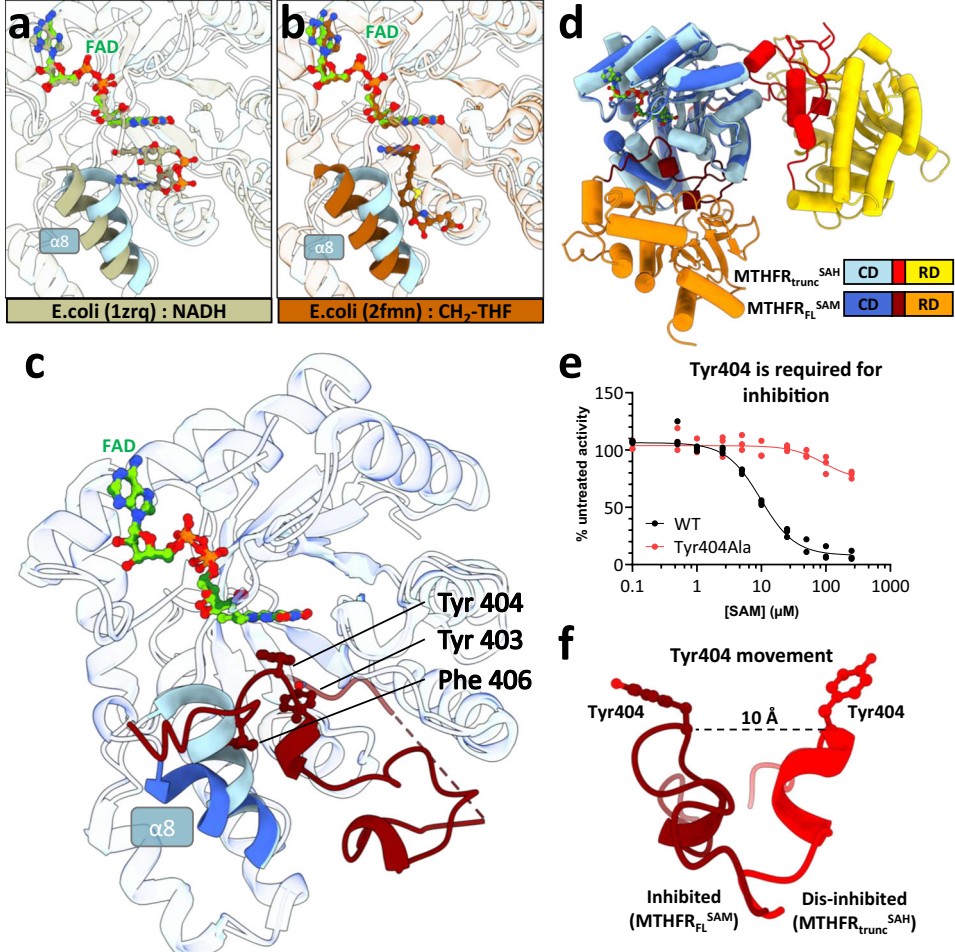

**Fig. 2 | MTHFR_FL^SAM reveals steric hindrance of the active site by hydrophobic linker insertion.** Zoomed view of the active site for MTHFR_trunc^SAH overlayed with crystal structures of *Escherichia coli* MTHFR homologues to highlight stacking interactions of FAD with NADH (**a** PDB: 1zrq[24]), and 5,10-dideazafolate analogue (LY309887) representing the position of 5,10-methylenetetrahydrofolate (CH₂-THF) in Ala177Val MTHFR background (**b** PDB: 2fmn[20]). **c** Overlay of the CD of MTHFR_trunc^SAH (light blue) and MTHFR_FL^SAM (dark blue). Note the movement of helix α8 to accommodate the hydrophobic insertion of linker residues Tyr403, Tyr404,

and Phe406 in MTHFR_FL^SAM (dark red). In **a–c** white ribbons represent complete superposition. **d** Tube cartoon representation of monomeric MTHFR_trunc^SAH and MTHFR_FL^SAM aligned with respect to the CD. **e** Relative enzymatic activity of MTHFR_FL protein following incubation with SAM. Wild-type (WT) and Tyr404Ala MTHFR_FL were overexpressed in HEK293T MTHFR knock-out cells and measured as lysates. N = 3 biological replicates. **f** Comparison of the position of Tyr404 in MTHFR_trunc^SAH (red) and MTHFR_FL^SAM (dark red) illustrating the side-chain rotation and -10 Å shift in position.

CH₂-THF binding site in a competitive manner. The essentiality of the steric inhibition by the key insertion residue Tyr404 is underlined by its substitution to Ala404, which ablated SAM-induced inhibition (Fig. 2e) but importantly did not compromise the maximal activity of the enzyme (Supplementary Fig. 9d). This implies that the insertion does not modify the active site per se; instead, the approximately 10 Å movement and rotation of Tyr404 (Fig. 2f) enables it to both plug the active site and fasten itself to the FAD, in a 'block-and-lock' fashion.

### Mobility of the linker drives the inter-domain reorientation

The transition of MTHFR from the open, dis-inhibited conformer (in the presence of SAH) to the closed, inhibited conformer (in the presence of SAM), resulting in the CD-RD re-orientation and steric insertion, is facilitated by an extensive rearrangement of the linker connecting the two domains. Based on the MTHFR_trunc^SAH crystal structure, the linker was expected to comprise aa 338–362[16]. With the determination of MTHFR_FL^SAM, we have redefined a more extended linker, comprising three adjoining segments (LS1: aa 338–380, LS2: aa 381–393, LS3: aa 394–412), which display significant mobility (Fig. 3a).

The first linker segment (LS1) was not possible to model in both MTHFR_FL^SAM protomers and one protomer of MTHFR_FL^SAH (asymm), indicating high intrinsic flexibility and disorder. In the MTHFR_trunc^SAH crystal structure, however, this region adopts an α-helix (α12) and stabilises the SAH in its RD binding pocket through a β-turn contributing Arg345 and Pro348 to abut the SAH homocysteine moiety, as well as through the connection between strand β9 and helix α12 to position Ala368 into proximity with the SAH sulphonium centre (Fig. 3b, c and Supplementary Fig. 11a, b).

The second (LS2) and third (LS3) linker segments were previously interpreted to constitute part of the RD, connecting α12 and α13 in the MTHFR_trunc^SAH structure[16]. In MTHFR_FL^SAM, however, these two segments undergo substantial flexing to a different trajectory (average Cᵅ-RMSD 10.87 Å). Composed of a 12-residue turn, LS2 swings from its position in MTHFR_trunc^SAH, where it interacts with its equivalent in the MTHFR_trunc^SAH homodimeric interface (Fig. 3b and Supplementary Fig. 11b, c), to a position of penetrating and remodelling the RD binding pocket in MTHFR_FL^SAM (Fig. 3d, e and Supplementary Fig. 11d), as described further in the next section. This segment harbours the motif F_384PNGRW_389, which is strongly conserved across all eukaryotes (Supplementary Fig. 10). To validate the

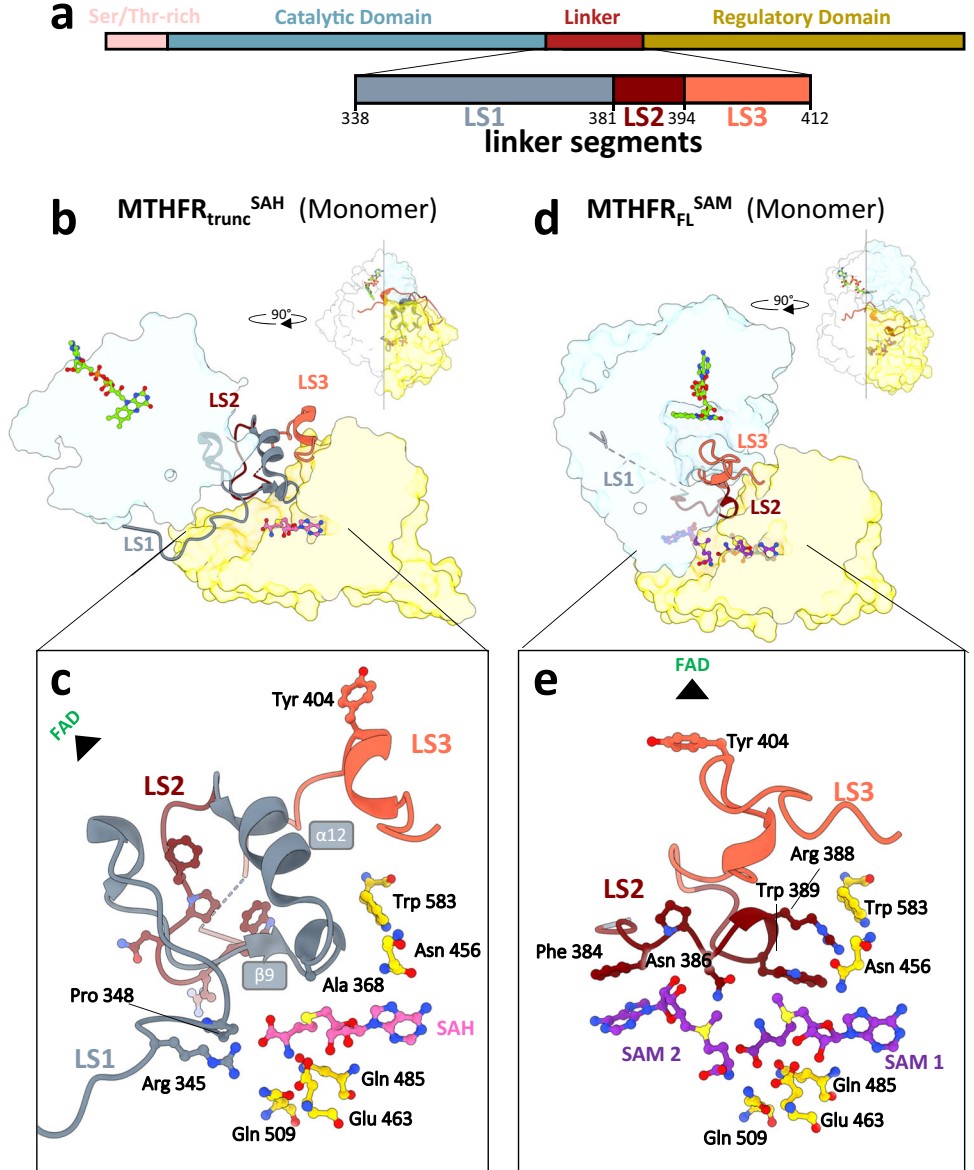

**Fig. 3 | Linker reorganisation drives inter-domain reorientation and remodels the allosteric pocket. a** Domain diagram of human MTHFR_FL, coloured according to MTHFR protein domains and with a zoom-in view of linker segments LS1 (aa 338–380), LS2 (aa 381–393) and LS3 (aa 394–412). The colour coding is maintained in subsequent structural representations. **b–e** Cut-through cartoon representation of monomeric MTHFR_trunc^SAH (**b**) and MTHFR_FL^SAM (**d**), showing the extensive rearrangement of LS1, LS2 and LS3. Insets show a more detailed and zoomed view of the respective allosteric pocket, containing one SAH for MTHFR_trunc^SAH (**c**) and two SAMs for MTHFR_FL^SAM (**e**).

importance of this motif, we performed alanine substitution of Phe384, Asn386, Arg388 and Trp389, each of which reduced enzymatic activity by 30–50% compared to wild-type MTHFR (Supplementary Fig. 11e). This underscores the necessity of the turn for fully functional MTHFR activity.

LS3 harbours the auto-inhibitory element Y_403YLF_406 including the insertion residue Tyr404. In MTHFR_trunc^SAH, this segment resides against the long helix α17 outside of the RD core (Fig. 3b and Supplementary Fig. 11b). In MTHFR_FL^SAM LS3 is drastically rearranged to adopt a position that packs closer to the CD and threads along the floor of the NADPH/CH_2-THF pocket in the CD (Fig. 3d and Supplementary Fig. 11d). This substantive displacement accounts for the long distance travelled by the insertion residue Tyr404 (10 Å), as well as the re-orientation of its aromatic sidechain (Fig. 2f), both of which are required for its steric blockage of the NADPH/CH_2-THF pocket.

## Remodelling of the RD pocket switches from binding one SAH to two SAM

The extensive conformational change of the linker region hugely impacts the SAM/SAH binding pocket of the RD. As anticipated, in the MTHFR_FL^SAM structure, we observed electron density corresponding to one SAM molecule (SAM1) within the RD pocket (referred hereafter as site 1) that was found to accommodate one SAH molecule in MTHFR_trunc^SAH (Fig. 3c, e and Supplementary Fig. 12a). The same RD residues that interact with SAH are employed in a similar manner to contact SAM1, including for example Glu463 and Thr573 (Supplementary Fig. 12b). However, the charged sulphonium centre enables SAM1 to establish an additional salt bridge with Glu463 (Supplementary Fig. 12b).

Immediately adjacent to SAM1, towards the verge of the RD (referred hereafter as site 2), an unexpected electron density was observed in MTHFR_FL^SAM (Fig. 3e and Supplementary Fig. 12a). This

density feature is not part of any contiguous protein density and initial automatic ligand placement by *Phenix-LigandFit* suggested it to be a second SAM molecule (SAM2). Both SAM1 and SAM2 molecules adopt the same extended configuration with respect to the methionine moiety, and *syn* configuration with respect to the nucleotide-ribose bond (Fig. 3c, e and Supplementary Fig. 12c). SAM1 and SAM2 are juxtaposed in a tail-to-tail manner with their adenosine rings pointing in opposite directions while their methionine moieties face each other (Fig. 3e). This juxtaposition allows hydrogen bonding between the methionyl carboxyl group of SAM1, the methionyl amino group of SAM2, and is further bridged by the strictly invariant residue Gln509 originating from the floor of the binding pocket (Supplementary Fig. 12b).

Our structures therefore reveal that the RD binding pocket undergoes major remodelling to accommodate either one SAH molecule (at site 1) in the dis-inhibited conformer ($MTHFR_{FL}^{SAH}$/ $MTHFR_{trunc}^{SAH}$ structures), or two SAM molecules (at sites 1 and 2) in the inhibited conformer ($MTHFR_{FL}^{SAM}$ structure) (Supplementary Fig. 12c). It is also apparent that site 2 is not present in the binding pocket of the dis-inhibited conformer, and only appears upon transition to the inhibited conformer. This switch in ligand binding mode and stoichiometry at the RD pocket is brought about by the substantive rearrangement between the two conformers, directed to the LS1 and LS2 segments of the extended linker region described in the previous section.

More specifically, in the presence of SAH, LS1 loops into the RD pocket to contact the homocysteine and sulphonium moieties, thereby adopting a position that 'seals off' site 2 (Fig. 3c and Supplementary Fig. 12c). Disorder in LS1 in the presence of SAM, by contrast, suggests that upon SAM binding this segment is displaced from the RD pocket to remove the seal. With site 2 open, binding of a second SAM molecule is possible and further stabilised by LS2, which swings from its position at the dimer interface (seen in the $MTHFR_{trunc}^{SAH}$ structure, Supplementary Fig. 11b, c) to the RD binding pocket (seen in the $MTHFR_{FL}^{SAM}$ structure, Fig. 3e and Supplementary Fig. 11d). The highly conserved $F_{384}PNGRW_{389}$ sequence (Supplementary Fig. 10) from LS2 plays a defining role in shaping site 2: Phe384 stacks against the adenosine ring of SAM2 (Fig. 3e and Supplementary Fig. 12b). Likewise, Arg388, Trp389 and Trp583 form a triple π-π stacking interaction in the presence of SAM2 (Fig. 3e).

In addition to contributions from the linker segments, site 2 is further shaped by the rearranged CD-RD orientation. As a result, the $D_{289}NDA_{292}$ sequence between helices α9 and α10 of the CD, which is oriented away from the dimer interface and exposed towards the exterior in the presence of SAH, traverses approximately 40 Å in the presence of SAM to be buried into the interior of the RD and to interact with SAM2 (Supplementary Fig. 11b, d). Asp289, Asp291, and Ala292 all interact with the adenosine group (Supplementary Fig. 12b).

### Both SAM binding sites are required for allosteric inhibition

Our structural data point to the unexpected binding of two SAM molecules in the RD pocket of each MTHFR protomer. Prompted by this two-SAM model, we searched for further evidence using biophysical and biochemical approaches. Isothermal titration calorimetry (ITC) of 500 μM SAM or SAH injected into 30 μM as-purified $MTHFR_{FL}$ revealed both ligands to bind the protein in an exothermic reaction, with stronger affinity exhibited by SAH ($K_d$ 0.38–0.39 μM) than SAM ($K_d$ 0.86–0.99 μM) (Fig. 4a and Supplementary Fig. 13a). Importantly, the binding isotherms revealed a binding stoichiometry for SAM ($N = 1.33$) that is double that for SAH ($N = 0.48$) (Fig. 4a and Supplementary Fig. 13a). Consistent with a more extended conformation than in the presence of SAM, SAH binding elicited a greater change in enthalpy as measured by ITC (Supplementary Fig. 12a). Furthermore, the SAM-bound conformer shows a higher melting temperature in DSF measurements (Supplementary Figs. 7f and 8a–d).

The two-SAM model would also imply that binding of both SAM molecules is necessary to elicit allosteric inhibition. To test this, we generated MTHFR variants in three systems: over-expression of variant MTHFR proteins in 293T cells with a genetic *MTHFR* knock-out, genetically engineered 293T cells to endogenously express variant MTHFR proteins, and purified recombinant variants of $MTHFR_{FL}$. To examine disruption of SAM1 binding residues we generated the variants Thr573Ala/Val, Gln485Ala/Leu and Glu463Gln; for SAM2 binding residues we generated Phe384Ala and Asn386Ala. To assess the importance of the triple π-π stacking interaction towards site 2 we also generated Trp583Ala, Trp389Ala, and Arg388Ala/Glu/Gln. ITC to evaluate binding of SAM to $MTHFR_{FL}$ variant proteins disrupted at site 1 (Glu463Gln) or the π-π stacking interactions (Arg388Gln) revealed no binding event upon injection of SAM to either variant protein (Supplementary Fig. 13b). Indeed, further examination of the enzymatic activity of variants engineered to disrupt the π-π stacking interactions as well as binding to site 1 and site 2 revealed abolished SAM-mediated inhibition in all systems tested (Fig. 4b, c and Supplementary Fig. 14a–j). Therefore, binding of SAM to both site 1 and site 2 is an integral part of this inhibitory process.

The requirement of both sites 1 and 2 further implies that they serve different roles in the inhibitory mechanism, both of which are required and will be achieved only above a certain threshold concentration of SAM (Supplementary Fig. 14k), ensuring displacement of SAH and occupation of both sites. It is therefore tempting to speculate that a single bifunctional ligand, with the capability to engage both sites 1 and 2, could mimic the potency of SAM-mediated inhibition at a potentially lower ligand concentration. We previously identified the SAM derivative (S)-SKI-72 (Fig. 4d) to inhibit MTHFR with a higher potency than SAM itself ($IC_{50}$ 0.8 μM vs 5.8 μM)[18]. We docked (S)-SKI-72 onto the $MTHFR_{FL}^{SAM}$ structure (Fig. 4e, Supplementary Fig. 15a–c)[18], showing that the ligand can engage with both sites 1 and 2, while also potentially navigating the space around the CD-RD linker region. Strikingly, the activity of $MTHFR_{FL}$ variants harbouring alanine substitution at residues contributing to binding site 1 (Glu463Gln) or the π-π stacking interactions (Arg388Gln) were inhibited by (S)-SKI-72 to a significantly diminished extent compared to that of wild-type $MTHFR_{FL}$ protein (Fig. 4f). Therefore, our data suggest that (S)-SKI-72 exerts its mode of inhibitory action in a similar manner as SAM, by being able to execute the roles, at least in part, designated for sites 1 and 2.

## Discussion

When we reported the crystal structure of human MTHFR, revealing a two-domain architecture with a novel RD fold appended to a conserved CD[16], we captured the enzyme bound with SAH which antagonises the inhibitory action of SAM. That structure adopted a conformation consistent with the role of SAH in dis-inhibiting the enzyme, such that the RD did not crosstalk with the CD. We postulated that crosstalk would occur when SAM binds to the same RD, transmitting a signal to the CD through inter-domain conformational changes. Now, through cryo-EM reconstruction of SAM-bound MTHFR, we have come to realise that the linker is at least 50 aa longer than originally designated, and crucially involved in this regulatory inter-domain crosstalk. Indeed, the direct and indirect participation of the extended linker defines the inhibitory mechanism of MTHFR, in three ways.

Firstly, the linker constitutes the multiplexed signal, transmitted through elaborated conformational gymnastics within its three constituent segments (LS1-LS3). In the presence of SAM, LS1 is displaced from the RD to the exterior, LS2 translocates from the dimer interface to inside the RD, while LS3 moves into the CD. In concert, these movements result in a substantial CD-RD re-orientation, akin to the closing of an oyster shell (Fig. 5a). Decoding this signal further, the LS1 and LS2 movements reshape the RD pocket to bind SAM, and the LS3

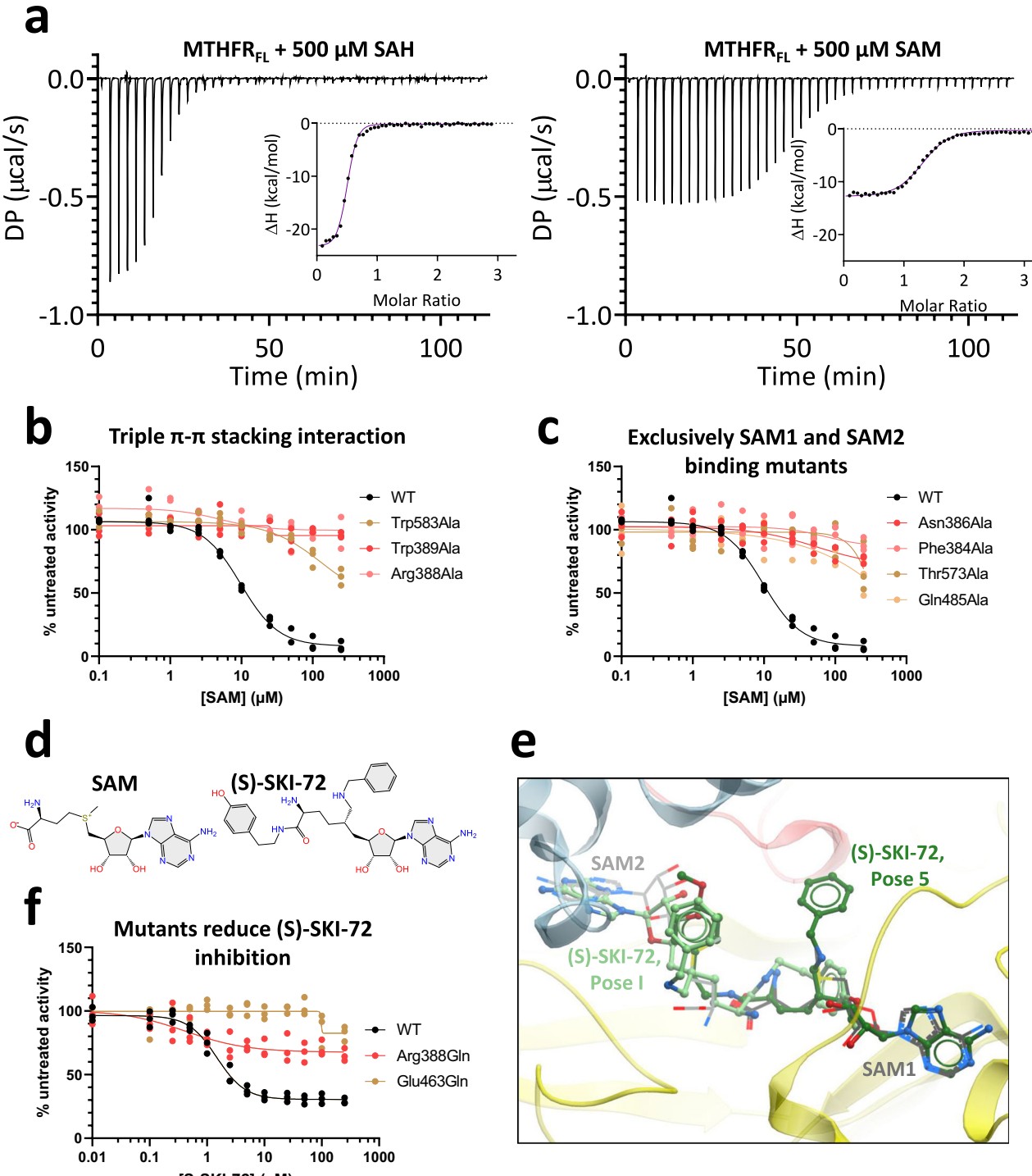

**Fig. 4 | Biochemical evidence supporting the two-SAM model. a** Representative ITC experiments displaying the heat exchange over time for the injection of 500 μM SAH or SAM into purified recombinant MTHFR$_{FL}$. DP: power differential. The inset illustrates ΔH (change in enthalpy) plotted against the molar ratio. All curves have been normalised against buffer injected into MTHFR$_{FL}$. $N = 2$ technical replicates, where the second replicate is shown in Supplementary Fig. 13a. **b** SAM inhibition curves of MTHFR$_{FL}$ and alanine substitutions of residues partaking in triple-π-π stacking interactions (Trp583, Trp389, Arg388) (**b**) or those which exclusively interact with site 2 (Asn386, Phe384), or site 1 (Thr573, Gln485) (**c**). The MTHFR$_{FL}$ variant proteins were overexpressed in HEK293T MTHFR knock-out cells and measured with an high-performance liquid chromatography (HPLC)-based

activity assay. The WT in (**b**) and (**c**) corresponds to the WT seen in Fig. 2e. $N = 3$ biological replicates. Maximum activity for each variant with corresponding Western blots are in Supplementary Fig. 14c, d. **d** Chemical structures of SAM and (S)-SKI-72. **e** Representative docking poses where (S)-SKI-72 adenine group occupies similar space to that of either SAM1 (Pose 5, dark green) or SAM2 (Pose I, light green) and extends towards the other SAM binding site. SAM molecules are shown as grey lines and (S)-SKI-72 poses are shown as green sticks. All docked poses and docking scores can be found in Supplementary Fig. 15a–c. **f** SAM inhibition curves for purified recombinant WT as well as Arg388Gln and Glu463Gln MTHFR$_{FL}$ protein incubated with (S)-SKI-72. $N = 3$ technical replicates.

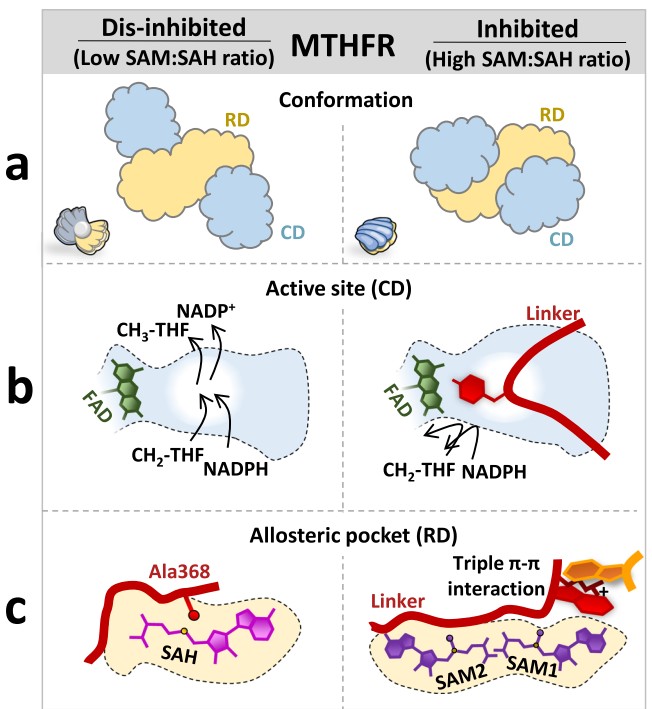

**Fig. 5 | Schematics summarising the allosteric regulation of MTHFR.** Each panel provides an overview of the **a** conformation, **b** active site and **c** allosteric pocket of dis-inhibited (left) and inhibited (right) MTHFR. Left, in the presence of a low SAM:SAH ratio, MTHFR is in an open conformation (like an open oyster), with free access to its active site and with only one SAH-bound in its allosteric pocket. Right, in the presence of a high SAM:SAH ratio, MTHFR is in its closed conformation (like a closed oyster) with the linker blocking the access to the active site, and two SAM molecules bound within the allosteric pocket.

movement directly results in active site inhibition. This multiplexed signal now offers the explanation as to why linker residues (particularly LS1 and LS2) directly influence the $K_M$ of NADPH in patients with MTHFR deficiency[27,28] and the sensitivity of SAM inhibition of yeast *met13*[29]. Inspection of MTHFR sequences indicates strong conservation of LS1 between mammals, plants and *C. elegans*, and LS2 (particularly F384PNGRW389 motif) across all eukaryotes, pointing to a universal mechanism of the SAM-bound signal transmission for MTHFR.

Secondly, the linker creates an allosteric inhibition mechanism which, in comparison to ligand induced activation mechanisms commonly employed by multi-domain metabolic enzymes, is far more intricate. In well-characterised examples of activation (e.g. CBS[30], PAH[31], DAH7PS[32]), the ground-state enzyme has the access to its active site partially hindered by a juxtaposed regulatory module, resulting in low basal activity; upon binding the activating ligand, this module dimerises and is sequestered away from the active site, allowing maximal activity. MTHFR has clearly evolved a completely different allosteric mechanism in response to an inhibitory (and not activating) ligand, which does not involve a dimerised element. Instead, MTHFR not only juxtaposes LS3 atop the CD, hindering access to the active site ('*block*'), but also plunges its aromatic Y403YLF406 motif into the CD, forcing away helix α8 and securely fastening itself to the cofactor FAD within the active site ('*lock*') (Fig. 5b). The complete loss of SAM-mediated inhibition by Tyr404 substitution is a testament to LS3's key role in this block-and-lock strategy.

Interestingly, the LS3 inhibitory sequences, including Y403YLF406 (and the CD helix α8 sequences) involved in the human block-and-lock strategy, are not preserved in lower eukaryotes e.g., *S. cerevisiae* MET13 and its homologue MET12[16]. We harnessed the predictive power of

Alphafold2[33,34] and surveyed the public repertoire of MTHFR models from lower eukaryotes. Strikingly, several models (e.g. *S. cerevisiae* Met12, https://alphafold.ebi.ac.uk/entry/P46151) adopt an inhibited conformation close to our experimental MTHFR$_{FL}$$^{SAM}$ structure, and similarly employ an aromatic (Phe) or hydrophobic (e.g. Ile, Leu) residue to block the NAD(P)H/CH$_2$-THF binding site. Altogether, we reason that adaptation of the block-and-lock inhibitory strategy for eukaryotic MTHFR was likely an early evolutionary event, although the exact sequence involvement could have diversified in higher eukaryotes.

Thirdly, and perhaps most unexpectedly, the linker directly participates in the creation of a second SAM binding site in the RD (Fig. 5c). Sealed off by LS1 in the SAH-bound dis-inhibited conformer, this second site (site 2) was only observed to be occupied by SAM when the adjacent site (site 1) was also SAM- (and not SAH-) bound. The location of site 2 in the MTHFR$_{FL}$$^{SAM}$ structure, more exposed to the exterior than site 1, now helps demystify the results of historic photoaffinity labelling studies[12,35] that identified residues in site 2 as important for SAM binding, an observation that puzzled researchers in the field for decades as these residues are some 50 amino acids N-terminal to the then predicted RD. Dual SAM binding has been previously reported for the radical SAM enzyme HemN in *E. coli*[36], where the two SAM ligands have non-equivalent binding sites and likely carry different roles; and *A. thaliana* threonine synthase[37], carrying two symmetrical SAM binding sites. As for MTHFR, the two SAM molecules bound to sites 1 and 2 are non-symmetrical in their ligand conformation and interaction, prompting us to also speculate that SAM1 and SAM2 coordinate different aspects of the inhibitory mechanism.

Since site 2 formation necessitates site 1 to be occupied by SAM and not SAH, we reason that site 1 is tasked with a sensor role, differentiating between the binding of SAM and SAH. The chemical structures of SAM and SAH differ by only a methyl group at the sulphonium centre. When site 1 is occupied by SAH (MTHFR$_{trunc}$$^{SAH}$ structure[16]), the demethylated sulphonium centre is 3.4–3.6 Å away from Ala368 (within LS1). When SAM1 binds to site 1, adopting an essentially identical ligand configuration to SAH (as shown in MTHFR$_{FL}$$^{SAM}$), the sulphonium methyl of SAM1 cannot be accommodated by Ala368 were it to retain the dis-inhibited MTHFR$_{trunc}$$^{SAH}$ conformation. We therefore speculate that the first event upon SAM1 binding is the repositioning of Ala368 to kick-start the LS1 conformational change, consistent with our original attribution of this residue as a SAM sensor[16].

While site 1 may function as a SAM/SAH sensor to trigger the initial process of SAM-mediated inhibition, site 2 likely serves the role of sustaining the inhibition process, further committing the protein to the LS1-LS3 conformational changes. To perform this role, the arrival of SAM2 removes the seal (formed by part of LS1) to site 2 and stabilises the LS2 triple π-π interactions (of Trp389, Arg388 and Trp583) within the RD core. These changes provide the platform of conformational change that ultimately leads to the LS3 block-and-lock. Our proposed dichotomous binding of SAM to the RD is consistent with the very slow (minutes) inhibitory mechanism of SAM[11] due to a tertiary conformational change from an active R state to the inactive T state restricting the affinity for NADPH binding[38]. However, the biphasic kinetics of the R to T[38] and the discovery of two SAMs further implies a possible 'poised' state of MTHFR that is singly occupied by SAM1, adopting a conformational transition between those captured by the MTHFR$_{trunc}$$^{SAH}$ and MTHFR$_{FL}$$^{SAM}$ structures. This putative state has so far evaded experimental entrapment. The existence of other ligand binding states (e.g., site 2 occupied by SAM/SAH with site 1 vacant), while unlikely, also remains to be clarified.

Without doubt, our data beg the question as to why the evolutionary adaption of MTHFR led to the assembly of three advanced modules (dual SAM binding, extensive linker rearrangement, and block-and-lock inhibition) into one metabolic enzyme. We are not

aware of any other multi-domain proteins that are constructed in this manner. There are over 18-fold types known to bind SAM[21], and SAM is a well characterised allosteric activator[30] and orthosteric inhibitor[39] of other human proteins. So, what is the imperative for MTHFR to evolve such a unique fold and mechanism, and not merely adapt from a plethora of already existing domains and mechanisms?

Perhaps the answer lies in the critical role that MTHFR plays in one-carbon metabolism, at the juncture of the folate and methionine cycles. MTHFR directly influences the rates of both SAM and pyrimidine biosynthesis in the cell, with important consequences for global trans-methylation and DNA synthesis[7]. To the best of our knowledge, MTHFR is the only human enzyme that is allosterically inhibited by SAM. The opportunity costs are likely to be so high for both inhibition and dis-inhibition of MTHFR, that a simple 'on/off' regulatory switch triggered by the mere presence/absence of SAM alone is not sufficient for cellular needs. Instead, a multi-layered switch with a built-in sensor that responds to the SAM:SAH ratio allows MTHFR to fine-tune its catalytic response to cellular needs. We propose that with a low SAM:SAH ratio in the cell, MTHFR adopts the dis-inhibited $MTHFR_{trunc}^{SAH}$ state to dedicate one-carbon flux to SAM biosynthesis. When the SAM:SAH equilibrium is tipped towards more SAM, its binding to site 1 primes the enzyme for inhibition. At a high SAM:SAH ratio, the excess SAM in the cell ensures subsequent binding of SAM to site 2, thereby triggering the cascade of events eventually leading to a fully inhibited protein (as seen in $MTHFR_{FL}^{SAM}$) and thereby a biological commitment to switch-off SAM biosynthesis.

This depiction, however, is still likely an over-simplification. MTHFR functions as a homodimer, where the two constituent protomers function essentially independently, and could therefore adopt different binding states. Considering this, a continuum of SAM-bound, SAH-bound and unliganded states of an MTHFR homodimer would more closely reflect the physiological scenario in the cell, in line with our previous observation of multiple MTHFR conformations from SAXS[16]. This is further coupled to the additional regulation mediated by Ser/Thr phosphorylation at the N-terminus of the enzyme. We have previously shown that this phosphorylation sensitises SAM inhibition by lowering its $K_i$[16]. Thus, it is likely that phosphorylation primes either SAM binding or the consequent conformational change. To do so, the N-terminal region would likely have to interact with one (or more) of the linker segments. In the $MTHFR_{FL}^{SAM}$ homodimer, LS1 from each protomer is disordered but would be exposed to the same face of the protomer as would the N-terminus, at the dimer interface. Despite the presence and likely phosphorylation of the N-terminal region in $MTHFR_{FL}$, it could not be resolved in our cryo-EM reconstruction due to intrinsic disorder. Therefore, future structural studies will be needed to clarify how the (de)phosphorylated N-terminus interacts with the rest of MTHFR and primes inhibition.

Beyond shedding light into the inhibitory mechanism, this study also illuminates distinctive strategies for the development of MTHFR inhibitors and activators, which could have a diverse set of therapeutic indications. Human MTHFR is a highly attractive target as one could minimise the risk of cross-reactivity by exploiting its unique fold within the human proteome. We envisage that small molecules targeting the $MTHFR_{FL}^{SAH}$ conformer, through binding to site 1, could be a therapeutic avenue for MTHFR deficiency where the residual enzyme could be dis-inhibited. On the contrary, compounds like (S)-SKI-72[18], which are potentially capable of fulfilling both SAM1 and SAM2 roles at the RD, could be developed as potent inhibitors, addressing disease states exhibiting MTHFR overexpression or hyperactivity that need to be down-regulated (e.g. cancer).

## Methods
### Cloning, expression, and purification
Cloning, expression, and purification of human full-length $MTHFR_{FL}$ wild-type and variants were generated as previously described for purified recombinant[16], overexpressed-[28,40] and endogenously expressed[40] MTHFR. Pure recombinant protein was expressed using baculovirus/insect cell system where pFB-CT10HF-LIC (#39191, Addgene) backbone harbouring $MTHFR_{FL}$ with a C-terminal flag/His$_{10}$ tag[16] and variants, generated with site-directed mutagenesis, were cloned in DH10Bac cells (#10361012, Thermo Fisher Scientific). Isolated bacmids were used for expression in SF9 cells (#12659017, Gibco) and protein was purified by affinity (Ni-NTA, Qiagen) followed by size exclusion (Superdex 200, #GE17-5175-01, Cytiva) chromatography. Purified MTHFR was concentrated to 10–15 mg ml$^{-1}$ and stored at −80 °C in storage buffer (50 mM HEPES pH 7.5, 500 mM NaCl, 5% glycerol, 0.5 mM TCEP). Overexpressed MTHFR were obtained by cloning pcDNA3-C-FLAG-LIC backbone (#20011, Addgene) harbouring $MTHFR_{FL}$ with C-terminal flag-tag and variants, generated with site-directed mutagenesis, which were then transfected using Lipofectamine 3000 (#15292465, Invitrogen) into 70–90% confluent HEK293T (CRL-3216, ATCC) MTHFR knock-out cells[40]. The cells were incubated at 37 °C for 72 h before harvested and snap frozen. Expression was confirmed by western blot. Endogenously expressed $MTHFR_{FL}$ and variants, were generated though homology-directed repair by mixing HEK293T (CRL-3216, ATCC) cells with Cas9 vector (Invitrogen, TrueCut Cas9 Protein v2, A36496), gRNA and ssODN template before treated with two pulses of 1150 V for 20 ms in a Neon transfection system (Invitrogen, MPK5000), followed by 48 h incubation before single cell seeded. All gRNA and ssODN can be found in Supplementary Table 2. The CRISPR/Cas9-edited mutants were validated by DNA extraction using QuickExtract solution (Lucigen, QER090150) followed by sanger sequencing prior to cell line expansion[28,38]. Site-directed mutagenesis was performed using Phusion High-Fidelity DNA Polymerase (New England Biolabs). All primer sequences can be found in Supplementary Table 3. Overexpressed and endogenously expressed $MTHFR_{FL}$ and variants were lysed by 30 min incubation with ice-cold lysis buffer (0.01 M K2HPO4-buffer pH 6.6 with 0.15% luberol). To prepare purified recombinant MTHFR for negative stain and cryo-EM imaging experiments, the flag/His$_{10}$-tag was removed by overnight incubation at 4 °C with His-tagged tobacco etch virus (TEV) protease followed by multiple washes on Ni-NTA loaded resin column with storage buffer (±20 mM Imidazole)[40,41]. Traces of imidazole were removed by buffer exchange and samples were concentrated before storage at −80 °C.

### Negative stain EM
TEV protease-cleaved purified recombinant $MTHFR_{FL}$ was diluted to a final concentration of 50 nM in buffer (20 mM HEPES pH 7.5, 200 mM NaCl) containing 5 mM SAM or 2.6 mM SAH. 3 μl sample was added to S160-3 Carbon Film 300 Mesh Cu (25) (Agar Scientific) grids, plasma treated using PELCO easiGlow™ (Ted Pella, Inc). The grids were blotted by hand, washed in ultra-pure water before stained using 2% (w/v) uranyl acetate. Imaging of the stained grids was performed using a Hitachi HT7800 microscope operating at 120 kV, located at the Electron Microscopy Research Services (EMRS) facility at Newcastle University. Images were captured on a Hitachi HT7800, at defocus with a magnification of 70 K and a pixel size of 1.8 Å. cryoSPARC-v4.2.1[42] was used for processing of micrographs.

### Cryo-EM sample preparation and data acquisition
All grids were glow discharged using PELCO easiGlow™ (Ted Pella, Inc) and blotted using Vitrobot (Thermo Scientific) using temperature 4 °C and 100% humidity for Quantifoil Au R1.2/1.3, 300 mesh grids and at 22 °C and 100% humidity for Au-Flat or UltrAuFoil R1.2/1.3, 300 mesh grids. For active $MTHFR_{FL}^{SAH}$, TEV protease-cleaved purified recombinant $MTHFR_{FL}$ was diluted to 2 mg/ml in EM-buffer 1 (20 mM HEPES, pH 7.5, 150 mM NaCl, 0.0025% Tween20) with 1 mM SAH or diluted to 1 mg/ml in EM-buffer 2 (20 mM HEPES, pH 7.5, 150 mM NaCl) with 1 mM SAH. For $MTHFR_{FL}^{SAM}$, TEV protease-cleaved purified recombinant $MTHFR_{FL}$ was diluted to 2 mg/ml in either cryo-EM buffer 2 or buffer 3

(10 mM Potassium-Phosphate, pH 6.6) with 5 mM SAM. All data collections were done at the York Structural Biology Laboratory (YSBL) on a Glacios equipped with a Falcon 4 direct electron detector (Thermo Fisher Scientific). EER formatted movies were imaged at 200 kV with a magnification of 240k, with a pixel size of 0.574 Å. Movies over 5.18 s were recorded with a defocus range of −0.9 μm to −2.1 μm with a total dose of 50 e⁻A². For MTHFR$_{FL}$$^{SAH}$ in total 5606 micrographs were collected and pooled from two different grids. For MTHFR$_{FL}$$^{SAM}$ in total 2394 micrographs were collected and pooled from three different grids. The two data sets were collected at two separate sessions. For further details see Supplementary Figs. 2a and 3a.

All datasets were imported into cryoSPARC-v4.2.1[42] and were subjected to patch motion and patch CTF correction. For more detailed information on the processing workflow for all datasets see Supplementary Figs. 2 and 3. In brief, for the MTHFR$_{FL}$$^{SAH}$ maps, a subset of micrographs prepared with EM-buffer 2 were subjected to blob picking and 2D classification. The 2D classes were then used for template picking for both micrograph sets prepared with EM-buffer 1 and 2, and individually processed and refined with C1 symmetry applied before pooled. Masking and C2 symmetry were applied to find correct symmetry axis for the MTHFR$_{FL}$$^{SAH (symm)}$ map. To determine the asymmetric SAH-bound state (MTHFR$_{FL}$$^{SAH (asymm)}$), particles from a C2 local refinement of the regulatory domain dimer were subjected to symmetry expansion. This was then followed by 3D classification in CryoSPARC to classify the heterogeneity of the flexible catalytic domains while maintaining the angles from the C2 symmetry applied local refinement, like symmetry relaxation as implemented in Relion[43]. Two sets of classes were determined representing the CD being less flexible for one protomer and absent for the other. One set of these classes were pooled and then subjected to a local refinement without symmetry applied. This used a soft mask that encompassed both the central RD dimer along with a single, less flexible, catalytic domain. The resulting map was then sharpened based off local resolution in cryoSPARC to aid in interpretability. Local masked refinements of the CD either resulted in over fitting or volumes that did not show any improvement. This is likely due to the size of the CD (approximately 38 kDa), its relative flexibility to the central RD dimer, and the orientation bias of the sample. For the MTHFR$_{FL}$$^{SAM}$ map processing, micrographs from Au-Flat grids prepared with EM-buffer 2 were first subjected to blob picking, ab initio and hetero refinement, where the best class representing dimeric MTHFR was further refined, re-centred and particles re-extracted with new coordinates. 2D classes from the new particle stack were used in template picking for all the three data sets, and refined by multiple rounds of heterogenous refinement, local motion correction with re-extraction, non-uniform refinement and local refinement. Overall, our data collections resulted in a map of the central RD dimer bound to SAH at 2.8 Å, an asymmetric map of the SAH-bound state at 3.1 Å, and a map of the SAM-bound state at 2.9 Å.

## Model fitting, refinement, and validation

For both MTHFR$_{FL}$$^{SAH}$ structures, the crystal structure of MTHFR$_{trunc}$ (PDB: 6fcx) was used as an initial model and rigid docked into both maps in ChimeraX[44]. Regions not represented in the density were appropriately deleted. For the asymmetric model the density of the CD is not well defined due to flexibility. Therefore, we docked a single CD guided by the 6fcx crystal structure into the unsharpened map and flexibly fitted using Namdinator[45]. Further refinement used the locally sharpened map. Sidechains of the CD were then truncated to alanine. For the MTHFR$_{FL}$$^{SAM}$ structure a homology model was created using SWISS-MODEL[46] and the AlphaFold[33,34] predicted structure of Met12 which appeared to be in an inhibited conformation and presented features like the MTHFR$_{FL}$$^{SAM}$ map. This homology model was docked into the density using ChimeraX[44] and regions of the model not represented by the density were deleted. SAM and FAD ligands were initially fitted using LigandFit in Phenix[47]. For refinement of all models

Namdinator[45] was used for flexible fitting followed by iterative rounds of ISOLDE[48], manual adjustment in COOT[49], and Phenix real space refinement[50]. Models were validated using MolProbity[51], model fit was assessed using the Q-score[52] implementation in ChimeraX[44], which was also used to create figures.

## HPLC activity assay

All enzymatic assays were performed with an HPLC-based physiological forward assay for MTHFR originally described by Suormal et al.[41] with adaptations for measurements in lysate from overexpression in HEK293T MTHFR knock-out cells or endogenous expression in genetically modified HEK293T[27,28], and in purified recombinant protein[16]. All enzymatic reactions were performed in 96-well plates and scaled down to reaction volumes of 20–50 μl with final reagent concentrations of 50 μM K2HPO4 buffer at pH 6.6, 100 μM of 5,10-methylenetetrahydrofolate (CH2-THF), 200 μM nicotinamide adenine dinucleotide phosphate (NADPH), 75 μM flavin adenine dinucleotide (FAD), and protein concentrations of 0.6 mg/ml for endogenously expressed in cell lysate, 2 μg/ml overexpressed in cell lysate and 32 ng/ml pure recombinant protein. Reactions were initiated by the addition of NADPH and CH2-THF, and incubated for 20 min (lysate) and 7 min (pure protein) 37 °C before stopped with 10–25 μl of 1% vitamin C in 5% HClO4. For ligand inhibition assay, the reaction mixtures were preincubated for 5 min at 37 °C with purified SAM[53], with SAH (#A9384, Sigma-Aldrich) or (S)-SKI-72[18] prior to reaction initiation. Final reaction products were diluted in 0.5% vitamin C solution before 5-methyltetrahydrofolate (CH3-THF) content was determined with HPLC (Jasco) equipped with nucleosil 120 C18 column (#720041.46, Macherey-Nagel). Curve fit was performed using inhibitor versus response [four-parameter curve fit] by GraphPad Prism (v9). Untreated specific activity was obtained from samples with no ligand addition. Raw data are available in Source data.

## Isothermal titration calorimetry

Purified MTHFR$_{FL}$ and variants were buffer exchanged using Zebra spin desalting columns (Thermo Scientific) using filtrated (0.2 μm) 25 mM HEPES buffer, with ligands diluted in the very same buffer. A MicroCal PEAQ-ITC microcalorimeter (Malvern Instruments, Malvern, UK) was used for all measurements, titrating ligand (500 μM SAM or 500 μM SAH) into the calorimetric cell containing MTHFR protein at approximately 30.0 μM or 10.0 μM. For runs of 45 injections, 0.8 μl injections were used, for 19 injections, 3 μl injection volumes were used. In both sets of experiments injections were spaced with 150 s intervals and ran at 25 °C. The resulting isotherms were subtracted against ligand titrated into buffer only runs and the data evaluated using MicroCal PEAQ-ITC software (Malvern Instruments). Raw data are available in Source data.

## Differential scanning fluorimetry

The binding events between SAM or SAH and MTHFR$_{FL}$ variants were detected either by the emitted fluorescence from a fluorometric dye or by detecting the emission from FAD. The samples were prepared in 96-well PCR plates, where each well (20 μl) contained 0.1 mg/ml purified recombinant protein, buffer (10 mM HEPES pH 7.5, 500 mM NaCl) and 1 mM SAM, 1 mM SAH or no ligand. The samples were incubated for 10 min at room temperature before the addition of SYPRO-Orange dye (#S6650, Invitrogen) diluted 500X or water for FAD emission. All fluorescence measurements were performed using QuantStudio 5 (Applied Biosystems), with SYBR set as the reporter, and a temperature ramp of 0.05 °C/s from 25 to 99 °C. Raw data are available in Supplementary Fig. 8a−d and Source data.

## Western blotting

Western blotting analysis was performed using as primary antibody 1:2000 diluted monoclonal mouse anti-flag (#F3165, Sigma) to target

MTHFR or 1:5000 diluted monoclonal mouse anti-β-Actin antibody (#A1978, Sigma) as loading control, and as secondary antibody 1:5000 diluted mouse IgG kappa binding protein conjugated to Horseradish Peroxidase (#sc-516102, Santa Cruz). Nitrocellulose membranes (#10600007, Cytiva) were developed using enhanced chemiluminescence detection reagents (#RPN2109, Cytiva, Amersham ECL Western Blotting Analysis System) and imaged with a ChemiDoc Touch Imaging System (Bio-Rad). Uncropped images are available in Source data.

### Docking of (S)-SKI-72

Docking was performed in ICM-Pro software (Molsoft LLC). To prepare the docking receptor, the MTHFR$_{FL}^{SAM}$ structure solved in this work was converted to an ICM object. After global energy minimisation, each protein chain was split into a separate object and SAM was removed from the model to clear the pocket. Next, receptor maps were made using the ICM interactive docking function, with the binding pocket defined as a box of $28 \times 28 \times 42$ Å, centred around both SAM molecules. (S)-SKI-72 was then converted to 3D, minimised and its adenine group was superimposed with the adenine group of either SAM1 or SAM2, as the starting position from which docking was performed. From each position, (S)-SKI-72 was docked into the prepared receptor, using ICM interactive docking function to retain the top 5 scoring binding poses (poses 1–5 from SAM1 starting point and poses I – V from SAM2 starting point). Poses were ranked according to three separate scoring functions: ICM-VLS score (a combined metric incorporating van der Waals interaction energy, number of torsions in the ligand, the solvation electrostatics energy change upon binding, the hydrogen bond energy, hydrophobic energy in exposing a surface to water, and the desolvation of exposed H-bond donors and acceptors); RT-CNN score (Radial Convolutional Neural Net including layers that do Topological (chemical graph) convolutions and 3D Radial convolutions; a neural network trained to recognise native-like complexes versus decoys directly, based only on geometries of putative complexes); and LE score (ICM-VLS score weighted to account for ligand strain).

### Reporting summary

Further information on research design is available in the Nature Portfolio Reporting Summary linked to this article.

## Data availability

Structures and EM maps have been deposited in the Protein Data Bank (PDB) and Electron Microscopy Data Resource (EMD) under the accession codes of PDB-8QA4/EMDB-18298 (MTHFR + SAH symmetric dis-inhibited state), PDB-8QA5/EMDB-18299 (MTHFR + SAH asymmetric dis-inhibited state) and PDB-8QA6/EMDB-18300 (MTHFR + SAM, inhibited state). Cryo-EM data have been deposited to the Electron Microscopy Public Image Archive (EMPIAR), EMPIAR-11959 (MTHFR + SAH) and EMPIAR-11926 (MTHFR + SAM). Structure not generated in this study, PDB-6FCX, PDB-1ZRQ and PDB-2FMN. All main data supporting the findings of this study are available within the article and Supplementary Information. Source data are provided with this paper.

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

## Acknowledgements

We thank Nicola Burgess-Brown and Eleanor Williams from Centre for Medicines Discovery University of Oxford, for the generous gift of the MTHFR expressing constructs. Negative stain electron microscopy was done at Newcastle University EM Research Services, and we are grateful to Tracey Davey for technical support. The Newcastle University EM Research Services is funded by the BBSRC (grant number BB/R013942/1). We thank Johan Turkenburg and Sam Hart for their assistance in collecting Glacios data at York Structural Biology Laboratory (YSBL). The YSBL is funded by the BBSRC, the Wellcome Trust (grant number 206161/Z/17/Z), Tony Wild, and the Wolfson Foundation. D.S.F. is supported by the Swiss National Science Foundation [310030_192505] and the University Research Priority Program of the University of Zurich ITINERARE – Innovative Therapies in Rare Diseases. L.K.M.B. was supported by the Children's Research Center Wifor Fellowship of the University Children's Hospital Zurich. W.W.Y. holds a visiting professorship at Nuffield Department of Medicine, University of Oxford.

## Author contributions

L.K.M.B., W.W.Y., D.S.F. and T.J.M. designed the experiments. L.K.M.B. performed protein expression and purification, collected EM data, analysed, and refined all MTHFR structures, along with the ITC, DSF, western blot, and activity assays. M.H. performed site-directed mutagenesis, protein over-expression, western blot, activity assays and assisted in CRISPR/Cas9 work. S.R.M. carried out in silico docking. C.B. designed and created CRISPR/Cas9 modified 293T cell lines. A.B. assisted in initial EM processing. T.J.M. assisted in screening, collecting, and processing EM data, along with ITC experiments. L.K.M.B., W.W.Y., D.S.F. and T.J.M. analysed the data and wrote the manuscript with contributions from all authors. W.W.Y., D.S.F. and T.J.M. supervised the project. W.W.Y. and D.S.F. conceived the project and procured funding.

## Competing interests

The authors declare no competing interests.
