## [Peer Review File · Nature Communications]

Dynamic inter-domain transformations mediate the allosteric regulation of human 5, 10-methylenetetrahydrofolate reductaseREVIEWER COMMENTS

Reviewer #1 (Remarks to the Author):

Overall Evaluation: Publish after minor revisions

In this excellent manuscript, Blomgren et al describe the determination of cryo-EM structures of human methylenetetrahydrofolate reductase (MTHFR) bound to the allosteric inhibitor S-adenosylmethionine (SAM) and to S-adenosylhomocysteine (SAH), which relieves SAM inhibition. The structures demonstrate an elaborate and truly breath-taking mechanism of allosteric regulation of hMTHFR. Besides illuminating the SAM/SAH regulation, the manuscript suggests effective strategies for inhibition of this important enzyme.

The manuscript is clearly written. The authors describe a very thorough structural study and effectively explain the results in the context of past work on hMTHFR. I recommend that the manuscript be accepted for publication in Nature Communications after addressing the following points.

1. Line 620 states that the HPLC-forward physiological assay for MTHFR is described by references 27, 28, and 16. I have checked these references and ref 27 states (p. 612) that activity was measured under saturating substrate concentrations (100 μ M methyleneTHF and 200 μ M NADPH). The saturating concentrations of substrates can change with different mutants. Thus, I suggest that the authors provide the saturating substrate concentrations for the various mutants measured in this study.
2. Figures 2b and Extended Figure 6b display the pdb 2FMN of E. coli MTHFR. Both figure legends and text (line 161) identify the ligand as CH₂-THF. However, as shown in ref. 20., this ligand is the 5,10-dideazafolate analogue, LY309887, obtained from Lilly, not CH₂-THF. Also, the enzyme is the Ala177Val mutant, not the wild-type E. coli MTHFR enzyme. I believe that the correct ligand/enzyme should be stated.
3. Line 1179-181 state that residues Lys270, Leu271, and Ser272 are conserved from mammals to worms and have been previously implicated in differentiating NADH (bacterial MTHFRs) vs. NADPH (eukaryotic MTHFRs) specificity. Ref 20 p. 11454 describes Lys 222 as possibly exerting long-range electrostatic effects. I believe that Lys270 of hMTHFR corresponds to Lys222 of eMTHFR, but it would be helpful to the reader to state this and/or include E. coli MTHFR in the sequence alignment in Supplementary Data Fig. 2.
4. Line 54 cites several references for regulation of MTHFR by SAM and SAH. The one key reference not cited is Jencks and Matthews, JBC (1987) 262, pp.2485-93, "Allosteric Inhibition of Methylenetetrahydrofolate Reductase by Adenosylmethionine: Effects of adenosylmethionine and NADPH on the equilibrium between active and inactive forms of the enzyme and on the kinetics of approach to equilibrium". While the current study demonstrates that the regulation by SAM/SAH is not a simple T to R transition, I believe it would be helpful to relate the structural findings to this early work by

Matthews.

5. Ref 26 is cited in line 180 as showing that NADH is the electron donor for bacterial MTHFR. I believe ref 24 should also be cited here where the differential specificity of NADH vs NADPH is discussed in the context of the E. coli MTFHR-NADH bound structure.

6. Extended Figure 6b has the E. coli residues switched. The lineup should be Thr94/Thr59 and Tyr321/275.

Thank you for allowing me the opportunity to review this manuscript.

Reviewer #2 (Remarks to the Author):

Blomgren et al described the cryoEM structure determination of methylenetetrahydrofolate reductase. This human flavoenzyme is of paramount importance in one-carbon metabolism and it is regulated by the balance between SAM and S-adenosyl-L-homocysteine (SAH). The structure beautifully unveils the conformational changes underlying activation and inactivation. The SAH-bound protein shows the active site in the conformation capable of binding the NADPH and folate substrates whereas the regulatory domain is flexible. The binding of two SAM molecules stiffen the protein in an inactive state where the residues of the domain linker occlude the active site. This structural mechanism for enzyme regulation was validated by the investigation of several mutants targeting the residues that are part of the regulatory regions. This is excellent work. The Authors have done an outstanding job with the narrative and the associated figures. The description of the conformational changes are of exemplary clarity. This work will be very relevant both in the fields of metabolism and basic enzymology. I have a few minor suggestions that might further improve an otherwise outstanding manuscript.

-Would it be possible to show how the activity as a function of varying SAM/SAH ratios e.g. using as reference a fixed SAH or SAM concentration of 100 nM or 1 micromolar? For instance, what is the activity at a 1:1 ratio?

-Related to the previous point, can the Authors remind the reader of the activity in the absence of any ligand?

-Would the structure of the unligated enzyme resemble the active or inactive conformation?

Reviewer #3 (Remarks to the Author):

In this manuscript, the authors detail the structural studies using cryoEM and complementary biochemical techniques to detail the contributions of the linker regions and a previously undocumented SAM binding site in the allosteric regulation of MTHFR. Although the authors are clearly experts in the field of MTHFR biology, there are concerns about the quality of the cryoEM, primarily regarding the asymmetric dis-inhibited complex, that raises serious concerns about some of the conclusions drawn. Although the symmetric SAH and SAM complexes appear to be of high quality, the same cannot be said for the asymmetric structure. My concerns are outlined below.

“MTHFRFL protein suffered from some orientation bias on grids, which we overcame by the application of C2 symmetry in refinements”

-The limitations of orientation bias in cryoEM structure determination are well-documented and efforts to overcome said issues are vast. Indeed, except for cases of icosahedral symmetry (e.g., AAV2 virus capsid), application of symmetry does not solve this issue (see Ayer et al. 2021 and Tan et al. 2017) and often leads to inflated resolution and unreliable densities. In cases of orientation bias, modifications of sample preparation and/or data collection (again see Tan et al. 2017) are necessitated. This is particularly important since symmetry relaxation was required for the C1 reconstruction of the asymmetric complex. Indeed, artifacts results from preferential orientation are present in both the C2 and C1 reconstruction outlined in Extended Figures 2e and h. As such, the features of the maps suffer from over sharpening, resulting in overfitted atomic models and lower-than-expected Q-scores, as evidenced in the provided plots. The elevated B-factors are also indicative of inflated resolution and over sharpening (except for the SAM structure). These observations limit the ability for the authors to make detailed claims on amino acid movements and distances with high confidence.

Following up on these findings, the CD cannot be confidently placed in the MTHFRtrunc structure. I displaced the CD randomly by 5 Å with random rotations and performed rigid body fitting in ChimeraX with inconsistent results (>2.5 Å RMSD). The lack of consistent placement and unstable refinement raises concerns about the rotation and distance measurements the authors claim here:

“This inter-domain rearrangement manifests as a -'34° rotation coupled to -'14.8 Å translation, long an axis that intersects with the linker region connecting the CD and RD” Cannot claim these values with the current quality of density.

“The CD-RD orientation and conformational asymmetry across protomers seen here agree with that of the SAH-bound crystal structure”

Again, cannot unambiguously claim this.

The lack of convergence during refinement and broken density in the CD region, in particular the lack of convincing density for many of the helices in the CD, including the aa233-246 helix, and missing FAD cofactor, are of concern. As evidenced by the local resolution plot, the drastic changes in estimated local resolution values indicate artifactual density that is unreliable.

Although the authors claim the flexibility and conformational dynamics limit resolution, local masking, and refinements, etc. should be able to rectify these issues to some extent. Yet, the authors do not appear to explore these options.

Furthermore, the authors used template picking for particle identification, but it is unclear from the text and figure if they used templates from 2D classification (not recommended) or from projections of a model. If the initial data are indeed limited in views, templates from 2-D classification will reinforce this bias. Have the authors employed processing strategies to use projections from low-resolution volumes (e.g. 20-30 Å low-pass filter) for picking with 2-D classification strategies to prevent template-based overfitting? The size of the CD is large enough that symmetry expansion should not be necessary as low-resolution features should drive alignment. My concern is that the weakened density for the CD is a mixture of data processing weaknesses, conformational flexibility, and compositional heterogeneity that have not been properly handled. Indeed, the falloff in density for the CD when thresholding EMD-18299 is consistent with non-stoichiometric occupancy and over-sharpening.

The authors mention the CD of one protomer in the SAH complex is disordered but do not speculate further on why an entire domain would be missing. For many samples, large conformational changes can be observed in ensembles, even if the domains are not large. It is concerning the sample exhibits preferential orientation and a domain is completely missing in the reconstruction.

The authors mention adding Tween-20 for one of the SAH datasets but do not show evidence it helped with views. Furthermore, the changes in pH and buffer conditions for the different datasets are not elaborated upon, particularly for the SAM complex which was collected at pHs 7.6 and 6.6. I'm assuming the authors explored these conditions to benefit populating different views, but the authors should explicitly comment on this.

The thresholding of the densities to show the SAH and SAM cofactors indicated close zoning at very low thresholds in ChimeraX. This most likely results from oversharpening of the maps and radiation damage of the cofactors. The latter is most concerning given the weakened density around the sulfonium centers.

I would implore the authors to re-approach the processing and refinement of the asymmetric structure to bolster the claims in the manuscript. An improved density for the CD would better support the authors hypothesis and solidify their conclusions.

Minor:

“hetro” typo in extended figure 2

“Tiple” in Figure 4b

REVIEWER COMMENTS WITH AUTHOR RESPONSES

Reviewer #1 (Remarks to the Author):

Overall Evaluation: Publish after minor revisions

In this excellent manuscript, Blomgren et al describe the determination of cryo-EM structures of human methylenetetrahydrofolate reductase (MTHFR) bound to the allosteric inhibitor S-adenosylmethionine (SAM) and to S-adenosylhomocysteine (SAH), which relieves SAM inhibition. The structures demonstrate an elaborate and truly breath-taking mechanism of allosteric regulation of hMTHFR. Besides illuminating the SAM/SAH regulation, the manuscript suggests effective strategies for inhibition of this important enzyme.

The manuscript is clearly written. The authors describe a very thorough structural study and effectively explain the results in the context of past work on hMTHFR. I recommend that the manuscript be accepted for publication in Nature Communications after addressing the following points.

1. Line 620 states that the HPLC-forward physiological assay for MTHFR is described by references 27, 28, and 16. I have checked these references and ref 27 states (p. 612) that activity was measured under saturating substrate concentrations (100 μ M methyleneTHF and 200 μ M NADPH). The saturating concentrations of substrates can change with different mutants. Thus, I suggest that the authors provide the saturating substrate concentrations for the various mutants measured in this study.

Response: We agree with the reviewer that saturating substrate conditions may be mutant dependent. Since we did not specifically assess this for each mutant, we have revised the text to indicate the specific conditions under which all proteins were assessed. These are described below and with revised text indicated on lines 647-651 (page 20) of the manuscript:

"All enzymatic reactions were performed in 96-well plates and scaled down to reaction volumes of 20-50 μ l with final substrate concentrations of 100 μ M of 5,10-methylenetetrahydrofolate (CH₂-THF), 200 μ M nicotinamide adenine dinucleotide phosphate (NADPH) and 75 μ M flavin adenine dinucleotide (FAD)."

2. Figures 2b and Extended Figure 6b display the pdb 2FMN of E. coli MTHFR. Both figure legends and text (line 161) identify the ligand as CH₂-THF. However, as shown in ref. 20., this ligand is the 5,10-dideazafolate analogue, LY309887, obtained from Lilly, not CH₂-THF. Also, the enzyme is the Ala177Val mutant, not the wild-type E. coli MTHFR enzyme. I believe that the correct ligand/enzyme should be stated.

Response: The reviewer is correct. We have made the following changes to the manuscript to clarify the genetic background and ligand used for this overlay.

Figure text of Fig. 2, lines 205-207, page 6:

*"...and 5,10-dideazafolate analogue (LY309887) representing the position of 5,10-methylenetetrahydrofolate (CH₂-THF) in Ala177Val MTHFR background (**b**, PDB: 2fmn²⁰)."*

Supplementary Fig.9, page 16:

"...and Escherichia coli MTHFR with Ala177Val background co-crystallized with 5,10-dideazafolate analogue (LY309887) representing the position of 5,10-methylenetetrahydrofolate (CH₂-THF) (PDB:2fmn, brown)."

3. Line 1179-181 state that residues Lys270, Leu271, and Ser272 are conserved from mammals to worms and have been previously implicated in differentiating NADH (bacterial MTHFRs) vs. NADPH (eukaryotic MTHFRs) specificity. Ref 20 p. 11454 describes Lys 222 as possibly exerting long-range electrostatic effects. I believe that Lys270 of hMTHFR corresponds to Lys222 of eMTHFR, but it would be helpful to the reader to state this and/or include *E. coli* MTHFR in the sequence alignment in Supplementary Data Fig. 2.

Response: We thank the reviewer for this suggestion. As shown in **Response Fig. 1** below, the alpha helix in human MTHFR (α Y, aa263-273) superimposes well with the *E. coli* structure counterpart, and we expect this structural conservation to be maintained across all domains of life, including prokaryotes. However, the amino acid sequences comprising this helix are distinct between eukaryotes and prokaryotes. For example, the equivalent position of human MTHFR residue Lys270 is not Lys222 in *E. coli*, but instead Asp225, based on the structural superposition. Since the structure, but not the sequence, is conserved, we prefer to exclude non-eukaryotes in our sequence alignment in Supplementary Fig. 10 page 17-18 (formerly Supplementary Data Fig. 2).

Response Figure 1. **Left:** Superimposition of alpha helix aa263-273 human MTHFR_{trunc}^{SAH}, with Lys270 indicated (light blue) and *Escherichia coli* MTHFR co-crystallised with NADH, with Lys222 indicated (PDB:1zrq, beige). **Right:** Sequence alignment shown in current Supplementary Fig. 10 (formerly Supplementary Data Fig. 2), with added *E. coli* MetF MTHFR sequence, highlighting amino acids 262-275 (human MTHFR numbering) and demonstrating a lack of amino acid conservation of *E. coli* MetF compared to eukaryotic proteins.

4. Line 54 cites several references for regulation of MTHFR by SAM and SAH. The one key reference not cited is Jencks and Matthews, JBC (1987) 262, pp.2485-93, "Allosteric Inhibition of Methylenetetrahydrofolate Reductase by Adenosylmethionine: Effects of adenosylmethionine and NADPH on the equilibrium between active and inactive forms of the enzyme and on the kinetics of approach to equilibrium". While the current study demonstrates that the regulation by SAM/SAH is not a simple T to R transition, I believe it would be helpful to relate the structural findings to this early work by Matthews.

Response: We agree with reviewer 1, the work of Jencks and Mathews from 1987 was indeed an early indication of the biphasic transition from active R state to inactive T state posed by the binding of two SAM molecules. Furthermore, their paper suggests the slow inhibitory effect of SAM was caused by a larger conformational change, which also aligns with our findings. We have therefore referred to the main findings from their article in our Discussion in lines 457-460 (page 14):

“Our proposed dichotomous binding of SAM to the RD is consistent with the very slow (minutes) inhibitory mechanism of SAM¹¹ due to a tertiary conformational change from an active R state to the inactive T state restricting the affinity for NADPH binding³⁸. However, the biphasic kinetics of the R to T³⁸ and the discovery of two SAMs further implies a possible ‘poised’ state of MTHFR that is singly occupied by SAM1, adopting a conformational transition between those captured by the MTHFR_{trunc}^{SAH} and MTHFR_{FL}^{SAM} structures.”

5. Ref 26 is cited in line 180 as showing that NADH is the electron donor for bacterial MTHFR. I believe ref 24 should also be cited here where the differential specificity of NADH vs NADPH is discussed in the context of the E. coli MTFHR-NADH bound structure.

Response: Reference 24 has now been added to previous line 180 (now line 187), page 5: *“...(electron donor in bacterial MTHFRs)^{24,26}*

6. Extended Figure 6b has the E. coli residues switched. The lineup should be Thr94/Thr59 and Tyr321/275.

Response: Thank you for catching this. We have adjusted the residue labels accordingly. See Supplementary Fig. 9 (formerly Extended Data Figure 6b), page 16.

Reviewer #2 (Remarks to the Author):

Blomgren et al described the cryoEM structure determination of methylenetetrahydrofolate reductase. This human flavoenzyme is of paramount importance in one-carbon metabolism and it is regulated by the balance between SAM and S-adenosyl-L-homocysteine (SAH). The structure beautifully unveils the conformational changes underlying activation and inactivation. The SAH-bound protein shows the active site in the conformation capable of binding the NADPH and folate substrates whereas the regulatory domain is flexible. The binding of two SAM molecules stiffen the protein in an inactive state where the residues of the domain linker occlude the active site. This structural mechanism for enzyme regulation was validated by the investigation of several mutants targeting the residues that are part of the regulatory regions. This is excellent work. The Authors have done an outstanding job with the narrative and the associated figures. The description of the conformational changes are of exemplary clarity. This work will be very relevant both in the fields of metabolism and basic enzymology. I have a few minor suggestions that might further improve an otherwise outstanding manuscript.

Would it be possible to show how the activity as a function of varying SAM/SAH ratios e.g. using as reference a fixed SAH or SAM concentration of 100 nM or 1 micromolar? For instance, what is the activity at a 1:1 ratio?

Response: We agree that demonstrating competition between SAM and SAH is interesting considering the findings of this paper. Now shown in Supplementary Fig. 14k page 24, we have compared the activity of MTHFR following application of 5 different concentrations (0, 0.1, 1, 10 and 100uM) of SAM against the same concentrations of SAH. The results demonstrate a reduced IC₅₀ for SAM with increasing SAH concentrations. We think this is line with the requirement of both sites 1 and 2 to be occupied with SAM for inhibition to take place, and further implies that these sites serve different roles in the inhibitory mechanism.

Figure reference and minor text adjustments, lines 344-347, page 10:

“The requirement of both sites 1 and 2 further implies that they serve different roles in the

inhibitory mechanism, both of which are required and will be achieved only above a certain threshold concentration of SAM (Supplementary Fig. 14k), ensuring displacement of SAH and occupation of both sites.”

Please note that we have additionally made minor recalculations to the inhibition assays, specifically in Figures: Fig. 4f, Supplementary Fig. 7a and Supplementary Fig. 14i. These have had no effect on the outcome nor interpretation.

-Related to the previous point, can the Authors remind the reader of the activity in the absence of any ligand?

Response: We are actually unsure of the activity of human MTHFR in the complete absence of ligand. Native MS studies of recombinant “as purified” MTHFR from SF9 insect cell expression indicated co-purification of either as a mixture of 0, 1 and 2 SAMs per dimer to phosphorylated MTHFR_{FL}, or 0, 1 and 2 SAHs to de-phosphorylated MTHFR_{FL} and MTHFR_{trunc} (Froese et al., Nat Commun 2018 (PMID: 29891918)). Therefore, to our knowledge, the activity of pure unliganded MTHFR has never been measured. In our hands, addition of SAH to “as purified” protein in either of these states has little effect on the maximum activity, which suggests that in the absence of any ligand the protein may have a similar activity as SAH-bound. To make this clear to the reader, we have added the following text in the manuscript:

Line 50-53 pages 1-2: “...whereby phosphorylated MTHFR has a reduced K_i for SAM and co-purify binding a mixture of 0-2 SAM molecules per protein unit¹⁶. Conversely, when MTHFR is de-phosphorylated, it co-purifies while binding 0-2 SAH molecules¹⁶.”

-Would the structure of the unligated enzyme resemble the active or inactive conformation?

Response: We thank the reviewer for this rather interesting question. We speculate that the unliganded enzyme would look most like the SAH-bound open state. However, we must emphasise that this speculation is entirely based on preliminary data we have of “as purified” MTHFR (without any ligand supplementation) and from existing literature. We have therefore intentionally refrained from discussing this state in the text as we need to investigate this further.

To this end, the preliminary cryoEM data set of “as purified” full-length MTHFR has been collected (ca 4000 micrographs) without any added ligand to the sample. However, the processing of the data is challenging. After various attempts, the best 2D class averages (of ca 7000 particles) we could obtain (**Response Fig. 2**) show poorly classified particles. We believe this can possibly be attributed to multiple factors: the flexibility of the catalytic domains, heterogeneity of the SAM stoichiometry, air-water interface denaturation of protein, and orientation bias of MTHFR particles. Further optimisation of this sample is needed for future studies.

Response Figure 2: 2D class averages of human MTHFR_{FL} as purified, prepared with 1mg/ml TEV-cleaved MTHFR_{FL} in 20 mM HEPES, pH 7.5, 150 mM NaCl, 0.0025 % tween-20, collected on Au Flat R1.2/1.3 300 mesh 1.0 mg/ml.

Furthermore, to investigate the true unliganded state of MTHFR, we have yet to establish a way to remove ligands from the allosteric pocket, such as dialysis, before attempting any structural work. This task, from our own experience, has not been trivial and should be addressed in future studies.

Reviewer #3 (Remarks to the Author):

In this manuscript, the authors detail the structural studies using cryoEM and complementary biochemical techniques to detail the contributions of the linker regions and a previously undocumented SAM binding site in the allosteric regulation of MTHFR. Although the authors are clearly experts in the field of MTHFR biology, there are concerns about the quality of the cryoEM, primarily regarding the asymmetric dis-inhibited complex, that raises serious concerns about some of the conclusions drawn. Although the symmetric SAH and SAM complexes appear to be of high quality, the same cannot be said for the asymmetric structure. My concerns are outlined below.

“MTHFR_{FL} protein suffered from some orientation bias on grids, which we overcame by the application of C2 symmetry in refinements”

-The limitations of orientation bias in cryoEM structure determination are well-documented and efforts to overcome said issues are vast. Indeed, except for cases of icosahedral symmetry (e.g., AAV2 virus capsid), application of symmetry does not solve this issue (see Ayer et al. 2021 and Tan et al. 2017) and often leads to inflated resolution and unreliable densities. In cases of orientation bias, modifications of sample preparation and/or data collection (again see Tan et al. 2017) are necessitated. This is particularly important since symmetry relaxation was required for the C1 reconstruction of the asymmetric complex. Indeed, artifacts results from preferential orientation are present in both the C2 and C1 reconstruction outlined in Extended Figures 2e and h. As such, the features of the maps suffer from over sharpening, resulting in overfitted atomic models and lower-than-expected Q-scores, as evidenced in the provided plots. The elevated Bfactors are also indicative of inflated resolution and over sharpening (except for the SAM structure). These observations limit the ability for the authors to make detailed claims on amino acid movements and distances with high confidence.

Response: We acknowledge the primary concern of the reviewer, that the asymmetric SAH bound map does not match the quality of the symmetric SAH and SAM complex maps. To make this more obvious to the reader, we now explicitly state the caveats of the asymmetric

map and model within our manuscript (Line 116-127, page 3). Further, based on the reviewer's reaction, we understand that we did not state clearly enough which models we have used to interpret the structural differences between the SAH and SAM bound states of MTHFR. Our interpretations of conformational changes are based off a comparison of our cryo-EM structure of the MTHFR_{FL} SAM bound state in the current study with our previously determined crystal structure of the SAH bound disinhibited state from using a slightly truncated construct (MTHFR_{trunc}^{SAH}, PDB code 6FCX). They do not rely on the asymmetric SAH bound structure. Although this lessens the burden for this latter map to be perfect, we nevertheless have taken the reviewer's comments extremely seriously. Below, find a sentence-by-sentence response to their comments.

1. "Application of symmetry does solve not this issue" (i.e. solve orientation bias):

To the manuscript, we have added a new figure of the resulting maps from local masked refinements of the central RD dimer bound to SAH (Supplementary Fig. 5, page 9) and associated text (Line 98, page 2). In both local refinements we used the same mask and the same particle stack, but in one refinement we applied C2 symmetry while in the other we applied no symmetry (C1). It is clear from visual inspection that secondary structure and side chain features appear only in the C2 symmetric local refinement. This is what we meant by the application of symmetry overcoming the effects of orientation bias. This is because applying C2 symmetry results in a map of the central RD dimer that has features agreeing with our previously determined crystal structure (MTHFR_{trunc}^{SAH}, 6FCX). To analyse this in more detail, we have taken advantage of the updated version of cryoSPARC released during the review process, and now have used the Orientation Diagnostics job on these maps and the others:

<https://guide.cryosparc.com/processing-data/all-job-types-in-cryosparc/utilities/job-orientation-diagnostics>

This new job measures both the conical FSC Area Ratio (cFAR) and the Sampling Compensation Factor (SFC*) of cryoEM maps and is based off Baldwin et al, Prog Biophys Mol Biol. 2020 (PMID: 31525386), Aiyer et al, Methods Mol Biol. 2021 (PMID: 33368004) and Tan et al, Nat Methods 2017 (PMID: 28671674) as referenced by the reviewer. We will not go into detail of what each of these parameters mean, as they are described in the cryoSPARC guide. Though both cFAR and SFC* consider particle alignments, the cFAR also considers the signal content of particles. For the local refinement without symmetry, we obtained a cFAR of 0.04 and SFC* of 0.63 whereas the C2 symmetric map has a cFAR of 0.24 and SFC* of 0.77 (Supplementary Fig. 5e, page 9). A cFAR value below 0.5 and an SFC* value below 0.81 are generally accepted as indications of preferred orientation which both maps clearly have. The map without symmetry applied is considerably poor. However, with the increase of both values and by visual inspection of the resulting map, we are confident that some of the issues due to orientation bias was overcome by applying C2 symmetry. Unlike the stated example of the HA trimer sample from Tan et al, Nat Methods 2017 (PMID: 28671674), applying C2 symmetry to the MTHFR sample overcomes some of the missing views in the C1 reconstruction for each RD protomer, as each particle has information for one Euler angle and its opposite for the regulatory domain monomer due to the symmetry (Supplementary Fig. 5d, page 9). Of course, if the orientation bias of the sample was different, i.e. if the bias was perpendicular to the symmetry axis where only one side view of one regulatory domain is seen, we would not expect to see such an effect from applying symmetry.

2. "...the features of the maps suffer from over sharpening, resulting in overfitted atomic models and lower-than-expected Q-scores, as evidenced in the provided plots.":

We agree with the reviewer's assessment that these maps have some over sharpening, but do not view that this results in overfitted models with lower-than-expected Q-scores. The C2 symmetric SAH bound map was sharpened with a Bfactor of 90.1 Å² determined from the

Guinier plot in cryoSPARC. With the preferred orientation bias, flexibility of the sample, and artefacts from applying C2 symmetry it is not surprising that over and under sharpening has occurred in some regions of the map. For example, the linker region is flexible and results in the artefacts seen in the C2 symmetric map (the disjointed density above the SAH molecule). Additionally due to the flexibility of the single CD and its much lower resolution in the asymmetric map we have locally sharpened this map based off the local resolution. We do state this in the Methods section of our manuscript, however we have changed the sentence to the following to make it clearer:

Lines 612-613, page 19: “*The resulting map was then sharpened based off local resolution in cryoSPARC to aid in interpretability.*”

This is the main deposited asymmetric map, but the sharpened map (based off a single Bfactor value of 86.3 \AA^2 determined from the Guinier plot for the local refinement for this map) and the unsharpened map are also available for inspection in the additional maps section for this deposition. The unsharpened map shows the presence of density for the CD (**Response Fig. 3**). Additionally, the unsharpened symmetric map is available in its EMDB entry. It is our expectation that if these maps were indeed over sharpened globally then the Bfactors would be much larger, typically above 300 \AA^2 , as observed when refinements overfit in cryoSPARC.

Multiple Views of the Unsharpened map of MTHFR_{FL}^{SAH} (asymm)
(EMD:18299) at a threshold of 0.0299 with model (PDB:8QA5)

Response Figure 3: Unsharpened map of asymmetric SAH bound MTHFR with PDB model.

Since we also believed that the CD of the asymmetric SAH bound MTHFR model could be overfitted, we based our modelling of the MTHFR_{FL}^{SAH} bound state on our previously determined crystal structure of MTHFR_{trunc}^{SAH}, 6FCX. (This is stated in the Methods section of our manuscript where we use 6FCX as an initial model for refinement.) An alignment of the appropriate domains of 6FCX against our models refined against the cryoEM maps gives low RMSD values of 0.536 \AA for the symmetric and 0.826 \AA for the asymmetric. Visual inspection of both models in the EM density show that side chains are in density that have sidechain features (**Response Fig. 4**).

Response Figure 4: Crystal structure 6FCX (light brown) aligned with model and sharpened maps (threshold: 0.09) from MTHFR_{FL}^{SAH(symm)} (light pink) and MTHFR_{FL}^{SAH(asymm)} (light blue), respectively.

With the CDs being so flexible in relation to the central RD dimer we expect that this part of the model for the asymmetric sample to be of lower quality/confidence than the rest of the molecule. This is why we have not modelled any side chains for this domain. However, as the CD is clearly present in both the 2D classes, the initial ab-initio map for these particles, and the unsharpened map, we felt that it is right to model the CD domain as backbone despite the caveats mentioned above and on Line 116-127, page 3 of the manuscript. We now explicitly state this in the Method section and give further details on the modelling:

Lines 629-632, page 19: *“For the asymmetric model the density of the CD is not well defined due to flexibility. Therefore, we docked a single CD guided by the 6FCX crystal structure into the unsharpened map and flexibly fitted using Namdinator⁴⁸. Further refinement used the locally sharpened map. Sidechains of the CD were then truncated to alanine.”*

To address the reviewer’s concerns over the Q-scores we also present the plot of average Q-score vs resolution from Pintilie et al, Nat Methods 2020 (PMID: 32042190), with our SAH bound models added and highlighted (**Response Fig. 5**). Our SAH models fall within the expected average Q-score range for their resolution from this plot: the symmetric map with a resolution of 2.8 Å has an average Q-score of 0.54 for both chains; the asymmetric map at a resolution of 3.1 Å has for chain A an average Q-score is 0.33 whereas for chain B its average Q-score is 0.42. The lower Q-score for chain A is due to the presence of the atomic model (backbone only) of the flexible catalytic domain. This is shown in our Q-score plots, and it was presented to make it clear to any reader that this region is of low quality (Supplementary Fig. 4b, page 7). As has been previously stated the local resolution and

quality of this region is far lower due to the intrinsic flexibility of catalytic domain relative to the central regulatory domain dimer (see **Response Fig. 8 below**).

Response figure 5: a, Average Q-scores vs. reported resolution for maps and models obtained from EMDb over only protein atoms modified from Pintilie et al, Nat Methods 2020 (PMID: 32042190). MTHFR_{FL}^{SAH(symm)} resolution: 2.8Å, average Q-score: 0.54(★). MTHFR_{FL}^{SAH(asymm)} resolution: 3.1Å, average Q-score: 0.36 (★), A chain Q-score: 0.33 (★) and B-chain Q-score: 0.42 (★).

3. “The elevated Bfactors are also indicative of inflated resolution and over sharpening (except for the SAM structure)”:

The Bfactors are elevated for the FAD and CD for the asymmetric map as is expected in this region due to the reasons we explained above and in the manuscript. Below we show figures of the atomic models coloured by their Bfactor both from our cryoEM maps and our crystal structure of MTHFR_{trunc}^{SAH} (6FCX) (**Response Fig. 6**). The crystal structure has an average Bfactor of 92 Å² for protein and 98 Å² for ligands. The Bfactor also varies across the model, with the RDs having the lowest Bfactors and the CDs the highest. Indeed, one CD of the homodimer has clearly higher Bfactors than the other. For this reason at the time we could not confidently model some regions of this CD due to disorder and flexibility (see Froese et al., Nat Commun 2018 (PMID: 29891918)). We believe this flexibility and asymmetry is also reflected (and enhanced) in our cryoEM structures of this state. For the symmetric map, the model has an average protein Bfactor of 29.7 Å² and ligand Bfactor of 29.5 Å². This is reflected in the figure of this model coloured by Bfactor (**Response Fig. 6**). For the asymmetric map the model has an average protein Bfactor of 92.8 Å² which is like both the symmetric model and our crystal structure. The ligand Bfactor is indeed inflated to 224.1 Å² but this is due to our decision to keep the FAD cofactor in the flexible, but present, CD (**Response Fig. 6**).

Response figure 6: Bfactors of MTHFR structures from crystallography and cryoEM.

4. “These observations limit the ability for the authors to make detailed claims on amino acid movements and distances with high confidence.”:

Any of our claims about amino acid movements and distances in terms of the CD are made by comparing our cryoEM derived model of the $MTHFR_{FL}^{SAM}$ bound state to our previous $MTHFR_{trunc}^{SAH}$ crystal structure. We apologise that this was not stated clearly enough in the previous version of our manuscript.

We hope the above explanations and figures have now alleviated the reviewer’s concerns about the limitations and our use of the asymmetric SAH bound structure.

Following up on these findings, the CD cannot be confidently placed in the $MTHFR_{trunc}$ structure. I displaced the CD randomly by 5 Å with random rotations and performed rigid body fitting in ChimeraX with inconsistent results (>2.5 Å RMSD). The lack of consistent placement and unstable refinement raises concerns about the rotation and distance measurements the authors claim here:

“This inter-domain rearrangement manifests as a $\sim 34^\circ$ rotation coupled to ~ 14.8 Å translation, long an axis that intersects with the linker region connecting the CD and RD”
Cannot claim these values with the current quality of density.

Response: We expect the reviewer is referring to our $MTHFR_{FL}^{SAH(asymm)}$ map, and not the $MTHFR_{trunc}^{SAH}$ which is the previously determined crystal structure (6FCX). Again, we would like to point out that the comparisons outlined in this comment were made in respect to the crystal structure (referred to as $MTHFR_{trunc}^{SAH}$) in the manuscript, not $MTHFR_{FL}^{SAH(asymm)}$. We fully acknowledge that this was not clearly stated in the text and have made the following change:

Lines 137-140, page 3: “By overlaying the $MTHFR_{FL}^{SAH}$ (cryo-EM) and $MTHFR_{trunc}^{SAH}$ (crystal) structures using the homodimeric RD interface (C α -RMSD 0.691 Å) (Supplementary Fig. 7d), we observed a significant rigid body movement of each CD relative to its own RD (Fig. 1b).”

“The CD-RD orientation and conformational asymmetry across protomers seen here agree with that of the SAH-bound crystal structure”

Again, cannot unambiguously claim this. The lack of convergence during refinement and broken density in the CD region, in particular the lack of convincing density for many of the helices in the CD, including the aa233-246 helix, and missing FAD cofactor, are of concern. As evidenced by the local resolution plot, the drastic changes in estimated local resolution values indicate artifactual density that is unreliable. Although the authors claim the flexibility and conformational dynamics limit resolution, local masking, and refinements, etc. should be able to rectify these issues to some extent. Yet, the authors do not appear to explore these options.

Response: As stated above, and in our submitted manuscript, we fully agree the density for the present CD in the asymmetric map is of lower resolution and poorer quality. This is because of the relative flexibility between the catalytic and regulatory domains (which we observed in our $MTHFR_{trunc}^{SAH}$ crystal structure and was also indicated by small-angle x-ray scattering, Froese et al., Nat Commun 2018 (PMID: 29891918)). However, the density for the CD is in the 2D classes, the ab-initio map, and unsharpened map for this sample (Supplementary Fig. 2, **Response Fig. 3**). We therefore found it most correct to include it in the final model. However, based on the reviewers concerns, we have now added more information when describing our analysis of the $MTHFR_{FL}^{SAH}$ structures (Page 3) to explain the caveats of this map and the models.

Lines 113-129, page 3: “This asymmetry was also seen in the ab-initio map during initial processing (Supplementary Fig. 2b). Here, the MTHFR homodimers appeared as asymmetric particles, where for one protomer both the linker and CD are highly disordered. In contrast, density was present for the linker and CD of the other protomer, though the density of both suggested flexibility (Fig. 1b,c and Supplementary Fig. 7c). It is important to note that the preferential orientation and possible denaturation at the air-water interface may explain the missing CD. Therefore, due to these issues, our modelling of this map is biased against our previously determined crystal structure of $MTHFR_{trunc}^{SAH}$ (see Methods). This SAH-bound crystal structure shows asymmetry in terms of the CDs where one is partially disordered and more flexible than the other¹⁶. Small-angle x-ray scattering (SAXS) also suggested that SAH bound MTHFR is more flexible in solution likely due to flexibility of the linkers and CDs¹⁶. This flexibility, the approximate CD-RD orientation, and conformational asymmetry seen in our model agrees with that of the SAH-bound crystal structure (Supplementary Fig. 7b). All considered together, our findings are consistent with the notion that SAH renders the MTHFR protein to be active, whereby the CDs are highly flexible and provide unhindered access of the substrate CH₂-THF and electron donor NADPH to the active site for catalysis.”

We did originally attempt local refinements of the single CD, but the majority of these resulted in overfitting and none gave density that we thought suitable for deposition. We present one of the better local refinements of the catalytic domain below (**Response Fig. 7**). Here we used the 105K particles of the C2 local refinement of the RD dimer and subjected them to 3D classification without alignment into four classes. Class two of 33K particles was then subjected to a local refinement with a softly padded mask focused on the CD. The features of the RDs become very distorted due to the relative flexibility of the CD. Though the density of the CD has more features, artefacts are present due to over fitting (a clear concern of the reviewer and the reason we did not find it suitable for deposition in the first

place). The difficulty in obtaining a good map focused on the CD is not surprising due to its small mass of approximately 38 kDa. Additionally, it is flexibly attached to the RD dimer (with a larger mass of approximately 74 kDa). We have therefore added the following sentence to the Methods section of our manuscript:

Lines 613-616, page 19: “Local masked refinements of the CD either resulted in over fitting or volumes that did not show any improvement. This is likely due to the size of the CD (approximately 38 kDa), its relative flexibility to the central RD dimer, and the orientation bias of the sample.”

a 3D Classes (no align) on 105k Particles From C2 Local and Mask Used For

b Unsharpened Map (3.5 Å) of Local Refinement Attempt On Catalytic Domain with Docked Model (8QA5)

0% transparency

50% transparency

c Zoom In On Catalytic Domain

Response figure 7: Example of local refinement attempt on CD. **a**, 3D classes of a 3D classification without alignment on the 105k particles from the C2 local refinement of the SAH bound RD dimer. Class 2 of 33k particles was used for a local refinement on the CD using the padded mask presented. **b**, The resulting map (unsharpened) of the local refinement focused on the CD at a resolution of 3.5 Å. The PDB 8QA5 of MTHFR_{FL}^{SAH(asymm)} is docked in the density. **c**, Zoomed in views of the CD density with docked PDB 8QA5. Signs of overfitting are present.

Furthermore, the authors used template picking for particle identification, but it is unclear from the text and figure if they used templates from 2D classification (not recommended) or from projections of a model. If the initial data are indeed limited in views, templates from 2-D classification will reinforce this bias. Have the authors employed processing strategies to use projections **from low-resolution volumes (e.g. 20-30 Å low-pass filter) for picking with 2-D classification strategies to prevent template-based overfitting?** The size of the CD is large enough that symmetry expansion should not be necessary as low-resolution features should drive alignment. My concern is that the weakened density for the CD is a mixture of data processing weaknesses, conformational flexibility, and compositional heterogeneity that have not been properly handled. Indeed, the falloff in density for the CD when thresholding EMD-18299 is consistent with non-stoichiometric occupancy and over-sharpening.

Response: We used blob picking of a subset of micrographs using these particles in 2D classification to create templates for picking of all micrographs. To make this clearer we have added following text regarding data processing in the methods:

Lines 599-604, page 19: *“In brief, for the MTHFR_{FL}^{SAH} maps, a subset of micrographs prepared with EM-buffer 2 were subjected to blob picking and 2D classification. The 2D classes were then used for template picking for both micrograph sets prepared with EM-buffer 1 and 2, and individually processed and refined with C1 symmetry applied before pooled. Masking and C2 symmetry were applied to find correct symmetry axis for the MTHFR_{FL}^{SAH (symm)} map.”*

Lines 616-622, page 19: *“For the MTHFR_{FL}^{SAM} map processing, micrographs from Au-Flat grids prepared with EM-buffer 2 were first subjected to blob picking, ab-initio, and hetero refinement, where the best class representing dimeric MTHFR was further refined, re-centered and particles re-extracted with new coordinates. 2D classes from the new particle stack were used in template picking for all the three data sets, and refined by multiple rounds of heterogenous refinement, local motion correction with re-extraction, Nu refinement and local refinement.”*

Further, to address the concerns of the reviewer, we created 2D templates of our 6FCX crystal structure to use for template picking for the SAH datasets. This was done by using the MolMap function in chimeraX to produce a map at a resolution of 5.0 Å which was resampled on the box of our asymmetric map. This 6FCX map was then imported into cryoSPARC, filtered to 10 Å, recentered on its C2 symmetry axis, and then 2D templates were created. The reviewer will be pleased to note that in cryoSPARC, 2D templates used for particle picking and the micrographs are low pass filtered to 20 Å by default. Although this can be altered by the user, we tend to use these default values. Below, we compare the templates used with their best resulting 2D classes (**Response Fig. 8**). In the majority of the 2D classes there is only density for a single CD in the 2D classes and still many views are missing. Two of the 2D classes (panel b, highlighted) possibly show two CDs. However, ab-initio followed by heterogenous refinement with six classes resulted in volumes that had only one CD present (panel c). Further attempts did not change this. This is likely due to the following reasons (non-mutually exclusive): denaturation of one CD (at air-water interface),

orientation bias, and/or high flexibility of the missing CD. We have interpreted our maps based on the previous crystal structure that shows asymmetry between the protomers within the dimer, where one CD is partially disordered and more flexible than the other. The reviewer's comments have made us realise that this could equally be caused by denaturation of one CD at the air-water interface, which could also be an explanation for the preferred orientation of the sample. Therefore, we have added the following sentence to our manuscript:

Lines 117-119, page 3: *"It is important to note that the preferential orientation and possible denaturation at the air-water interface may explain the missing CD."*

Response figure 8. Reprocessing of SAH bound dataset using 6FCX crystal structure as template for particle picking. **a**, 2D templates/projections of 6FCX used for particle picking. **b**, 2D classes of approximately 557k particles after three rounds of 2D classification of an initial stack of approximately 2 million particles. 2D classes with two possible catalytic domains are highlighted orange. **c**, Six volumes from one round of ab-initio and heterogenous refinement of approximately 447k particles. Two 3D classes present features of MTHFR reflective of the 2D classes. Density for only one catalytic domain is present for both volumes.

As mentioned above, we do not share the reviewer's view that the CD is large enough to drive alignment. It is only approximately 38 kDa and is flexibly attached to the central RD dimer of approximately 74 kDa. Local masked refinement in cryoSPARC is recommended to align regions of approximately 150 kDa which is nearly the total mass of our MTHFR sample. Regions smaller than this can be difficult to process due to low signal:

<https://guide.cryosparc.com/processing-data/all-job-types-in-cryosparc/local-refinement/job-new-local-refinement-beta>

Symmetry expansion combined with 3D classification with no alignment was done to use the assigned angles and shifts of the central RD from the C2 masked local refinement and to determine symmetry breaking features such as the linker and CD. This is because a global refinement of the initial ab-initio volume without symmetry resulted in a map with the expected low-resolution features but no interpretable secondary structure. In our view, the fall-off of density and over sharpening for the CD in the asymmetric map is because the main map of this deposition is locally sharpened based off resolution. The unsharpened map, which is available in the deposition, clearly shows the presence of density that we interpret to be the flexible CD (**Response Fig. 3**). The presence of a single CD is also shown in the 2D classes, and the ab-initio volume presented in Supplementary Fig. 2 of our manuscript. As such we believe that any weakness in our interpretation is because we are limited by the state of the sample in the grids in terms of preferential orientation, intrinsic flexibility, and possible air-water interface denaturation.

We will deposit all datasets to EMPIAR, so that the interested reader will have a chance to check for themselves and come to their own conclusions. As acknowledged at the outset, we accept that the MTHFR_{FL}^{SAH} dataset is not ideal, and there is certainly room for sample optimisation. However, there are no guarantees that even a perfectly optimised sample would reduce orientation bias nor result in a map with two CDs. As such, and with the re-analysis prompted by the reviewer, we believe that we have processed and handled the presented data properly and to the best of our ability and have produced maps with the expected features of secondary structure, while acknowledging to the reader the most important potential issues and caveats.

The authors mention the CD of one protomer in the SAH complex is disordered but do not speculate further on why an entire domain would be missing. For many samples, large conformational changes can be observed in ensembles, even if the domains are not large. It is concerning the sample exhibits preferential orientation and a domain is completely missing in the reconstruction.

Response: We agree with the reviewer that we could speculate further considering not only the high flexibility but also the possible denaturation of the single missing CD at the air-water interface. As such we have now added the following sentence (which we have mentioned in an earlier response but is repeated here):

Lines 117-119, page 3: *"It is important to note that the preferential orientation and possible denaturation at the air-water interface may explain the missing CD."*

The authors mention adding Tween-20 for one of the SAH datasets but do not show evidence it helped with views. Furthermore, the changes in pH and buffer conditions for the different datasets are not elaborated upon, particularly for the SAM complex which was collected at pHs 7.6 and 6.6. I'm assuming the authors explored these conditions to benefit populating different views, but the authors should explicitly comment on this.

Response: We have now included processing and refinement of the three different individual data sets with SAM in Supplementary Fig. 6, page 11, with associated text on Lines 98-100 of page 2. The variation of combined buffer and grid conditions indeed have an impact on the orientation bias. For example, the 10 mM K-P-buffer pH 6.6, Au-Flat particles show mainly a single side view of MTHFR. However, none of the individual sets achieve the same resolution as the pooled data set. Furthermore, after orientation diagnostics made for the final deposited map and from reprocessing of individual data sets with SAM, the cFAR are between 0.02-0.05 compared to our final pooled dataset with a cFAR of 0.3.

The thresholding of the densities to show the SAH and SAM cofactors indicated close zoning at very low thresholds in ChimeraX. This most likely results from oversharpening of the maps and radiation damage of the cofactors. The latter is most concerning given the weakened density around the sulfonium centers.

Response: We believe this is due to the way we have zoned the density around these ligands in ChimeraX rather than over sharpening and radiation damage. We have therefore re-zoned these ligand densities and have updated the following figures for SAH, FAD and SAM respectively in Supplementary Figs. 7b on page 12, 9a on page 16, 12a on page 20.

I would implore the authors to re-approach the processing and refinement of the asymmetric structure to bolster the claims in the manuscript. An improved density for the CD would better support the authors hypothesis and solidify their conclusions.

Response: Once again, we would like to make clear that our claims over the conformational changes are based on a comparison of our cryo-EM structure of MTHFR_{FL}^{SAM} with our previous crystal structure of MTHFR_{trunc}^{SAH}, and not with MTHFR_{FL}^{SAH}. We hope this single fact alleviates much of the reviewer's concerns. So that this misconception is not carried out by the reader, we have now made this distinction clearer in our manuscript. The new cryo-EM structures of MTHFR_{FL}^{SAH} do agree well with the crystal structure of MTHFR_{trunc}^{SAH}. However, the SAH-bound sample show severe orientation bias and maps showed density for only one CD. Though this apparent asymmetry is also reflected in the crystal structure, the reviewer's comments have made us realise that the other CD could be denatured at the air-water interface, potentially explaining the orientation bias of that sample. We hope that our additional figures and explanations now describe more clearly how we processed the SAH-bound sample data along with its caveats.

Following the suggestions of the reviewer, we have also extensively reattempted processing, as outlined in the figures and explanations above. Unfortunately, none have resulted in a substantially better model. Although this is unfortunate, as we were optimistic that this additional work would enable a higher quality model, we now opine that this is not because of data processing weaknesses, but rather due to the nature of the sample.

Minor:
"hetro" typo in extended figure 2b

Response: These are fixed in Supplementary Fig. 2 page 3 (formerly Extended Data Fig. 2) and in Supplementary Fig. 3 page 5 (formerly Extended Fig.3)

“Tiple” in Figure 4b

Response: This is fixed in Figure 4b, *page 11*

REVIEWERS' COMMENTS

Reviewer #1 (Remarks to the Author):

The authors have addressed all concerns from my first review of their manuscript.

Reviewer #2 (Remarks to the Author):

I confirm the evaluation given at the first round of revision: this is excellent work. It clarifies the regulatory mechanism operating in a fundamental enzyme of our metabolism. The manuscript is very well written. The concerns and comments given by the Reviewers have been convincingly addressed.

Reviewer #4 (Remarks to the Author):

After thoroughly reviewing the manuscript by Blomgreen et al., the comments from reviewer number 3 regarding the cryoEM processing, and the authors' response to the comments, I believe the manuscript adequately addresses the reviewer's concerns. I share the reviewer's perspective that it would be feasible to conduct additional data processing to obtain further insights into the catalytic domain that disappears during asymmetric processing. However, based on my own experience, domains of such small molecular weight (38 kDa) may exhibit different responses to such processing techniques. I acknowledge that the authors have undertaken all feasible efforts to improve the density of this domain and that its disappearance could be attributed to partial denaturation or significant heterogeneity due to its flexibility. The authors appropriately acknowledge these possibilities in their work, rendering the discussion sufficiently clear on this matter.

I have observed a small typo in line 99, in the word orientations.